# Small genomic insertions form enhancers that misregulate oncogenes

Brian J. Abraham[1], Denes Hnisz[1], Abraham S. Weintraub[1,2], Nicholas Kwiatkowski[1], Charles H. Li[1,2], Zhaodong Li[3,4], Nina Weichert-Leahey[3,4], Sunniyat Rahman[5], Yu Liu[6], Julia Etchin[3,4], Benshang Li[7,8], Shuhong Shen[7,8], Tong Ihn Lee[1], Jinghui Zhang[6], A. Thomas Look[3,4], Marc R. Mansour[5] & Richard A. Young[1,2]

The non-coding regions of tumour cell genomes harbour a considerable fraction of total DNA sequence variation, but the functional contribution of these variants to tumorigenesis is ill-defined. Among these non-coding variants, somatic insertions are among the least well characterized due to challenges with interpreting short-read DNA sequences. Here, using a combination of Chip-seq to enrich enhancer DNA and a computational approach with multiple DNA alignment procedures, we identify enhancer-associated small insertion variants. Among the 102 tumour cell genomes we analyse, small insertions are frequently observed in enhancer DNA sequences near known oncogenes. Further study of one insertion, somatically acquired in primary leukaemia tumour genomes, reveals that it nucleates formation of an active enhancer that drives expression of the *LMO2* oncogene. The approach described here to identify enhancer-associated small insertion variants provides a foundation for further study of these abnormalities across human cancers.

[1] Whitehead Institute for Biomedical Research, 455 Main Street, Cambridge, Massachusetts 02142, USA. [2] Department of Biology, Massachusetts Institute of Technology, Cambridge, Massachusetts 02139, USA. [3] Department of Pediatric Oncology, Dana-Farber Cancer Institute, Harvard Medical School, Boston, Massachusetts 02215, USA. [4] Division of Hematology/Oncology, Children's Hospital, Boston, Massachusetts 02115, USA. [5] Department of Haematology, UCL Cancer Institute, University College London, London WC1E 6DD, UK. [6] Department of Computational Biology, St Jude Children's Research Hospital, Memphis, Tennessee 38105, USA. [7] Key Laboratory of Pediatric Hematology & Oncology Ministry of Health, Department of Hematology & Oncology, Shanghai Children's Medical Center, Shanghai Jiao Tong University School of Medicine, Shanghai 200127, China. [8] Pediatric Translational Medicine Institute, Shanghai Jiao Tong University School of Medicine, Shanghai 200127, China. Correspondence and requests for materials should be addressed to A.T.L. (email: Thomas_Look@dfci.harvard.edu) or to M.R.M. (email: m.mansour@ucl.ac.uk) or to R.A.Y. (email: young@wi.mit.edu).

Tumour genomes can contain thousands of DNA variants that distinguish them from the genomes of healthy cells, including single-nucleotide substitutions, small and large insertions and deletions (INDELs), copy number alterations and translocations[1,2]. Only a small fraction of all variants, however, represent driver mutations that are truly pathogenic[3–5]. While the functions of numerous coding variants discovered in cancer cells through next-generation sequencing studies have been tested, the relevance of the numerous non-coding variants in the DNA of each human cancer remains largely unknown[5]. Few non-coding mutations have been investigated in depth, but among those studied, several play key roles in tumour biology, suggesting that non-coding drivers are underappreciated[6–10].

Non-coding variants that are potential drivers of tumour biology are likely to occur in gene regulatory elements, but their identification and verification can be challenging. For example, there is recent evidence that somatically acquired small INDELs can nucleate oncogenic enhancer activity[8], but this form of variation can be overlooked because sequencing technologies generally produce short reads that can be challenging to align to the reference genome[2,10,11]. The impact of non-coding variants within gene regulatory elements on oncogenic gene misregulation can be more challenging to establish than those that affect protein-coding sequences because gene regulatory elements are not as well defined and may occupy a larger fraction of the genome than protein-coding regions. To overcome these obstacles, several approaches have sought non-coding variants that alter transcription by incorporating gene expression and transcription factor motif position weight matrices into their discovery algorithms[12,13].

Here we propose an alternative strategy to identify bona fide non-coding driver mutations by analysis of sequencing reads from chromatin immunoprecipitation (ChIP-Seq) of the enhancer-associated histone mark H3K27ac (H3K27ac ChIP-Seq). This approach has an intrinsic advantage over whole-genome sequencing approaches to identifying functional variants because H3K27ac sequence reads are generated predominantly from active regulatory sites, providing a more direct link between the variant and putative function[14,15]. This approach dramatically reduces the search space and enriches for the set of variants that are likely to be functional at the level of gene control. We present a catalogue of enhancer-associated insertion variants from a panel of 102 tumour cell genomes and show they are frequently associated with known oncogenes. One example, a heterozygous 8 basepair (bp) insertion in T cell leukaemias proximal to the LMO2 oncogene, is demonstrated to affect gene control. This knowledge of enhancer-associated insertions provides a foundation for further studies to define the oncogenic contributions of this class of variants.

## Results

**Cataloguing enhancer-associated insertions.** To identify enhancer-associated variation in cancer cells and include insertion variants that are overlooked with common short-read alignment approaches, we developed a computational pipeline optimized to recover sequences of insertions that are present in tumor cells but are not present in the NCBI human reference genome (Fig. 1a, Supplementary Fig. 1A). The NCBI reference genome was used for comparison because most cancer cells do not have corresponding healthy samples for comparisons. The pipeline was used to analyse newly generated and previously published ChIP-Seq datasets for H3K27ac-enriched DNA from 78 tumour cell lines and 24 primary tumour samples, eight of which are new here (Supplementary Table 1)[8,16–43]. Using these enhancer-targeting ChIP-Seq datasets narrows the variant-

discovery search space to ~2% of each genome (Supplementary Fig. 1B). The computational pipeline was optimized to identify the subset of reads that could only be aligned to the reference genome when allowing for insertions in the reads, which were then analysed to discover the DNA sequence of the insertions themselves (Supplementary Fig. 1A). The pipeline leverages recent advances in alignment algorithms to permit the analysis of sequences that align only when allowing for the presence of insertions[11,44,45]. In addition, to aid in capturing somewhat larger (14–31 bp) insertions, contigs were assembled from initially unalignable reads and used in parallel with raw reads to reveal the underlying sequence (Methods). Although the majority of reads aligned without accounting for insertions (Supplementary Fig. 1C), there were a large number of insertions identified in these tumour samples. A catalogue of 328,871 candidate enhancer-associated insertions (Supplementary Data 1), which range in size from 1 to 31 bp (Fig. 1B, Supplementary Fig. 1D), were identified using this approach.

Although germline variation may contribute to oncogene misregulation[18], most cancer-driving variants are somatic in origin, so insertions judged likely to be background, germline variation were deprioritized for further study. Of 168,149 candidate enhancer-associated insertions with unique positions and/or sequences, 57,013 were deprioritized because they likely reflect germline variation based on two considerations, annotation and recurrence (Fig. 1c, Supplementary Data 1). Of the 57,013 putative germline insertions, 49,992 were present in dbSNP, which curates germline variants of multiple variant types across many databases[46]. Additionally, 20,715 variants were recurrent across multiple independent tumour types and samples and may thus reflect germline variation not represented in the reference genome. Indeed, 13,694 of the 20,715 nearly ubiquitous insertions were present in dbSNP[46], supporting the view that these insertions are predominantly germline. Thus, 111,136 (168,149 − 57,013) unique predicted enhancer-associated insertions appear, by these considerations, not to reflect germline variation and thus may be somatically acquired non-coding variants (Supplementary Fig. 1E). The catalogue of predicted enhancer-associated insertions described here thus expands the number of reported small insertions in enhancers of tumour cells from 10,165 (ref. 2) to 121,301 (10,165 + 111,136). The instances of enhancer-associated insertions observed in T cell acute lymphoblastic leukemia (T-ALL), breast, neuroblastoma (NB), lung, colorectal, melanoma, glioblastoma multiforme (GBM), B cell lymphoma (BCL), pancreatic, and other tumour cell types are summarized in Fig. 1d and Supplementary Fig. 1E. The pipeline was adapted to predict deletions in a similar manner (Supplementary Data 2), but we chose to focus on insertions for further study because of our previous experience with analysis of this class of variants[8].

**Confirming predicted enhancer-associated insertions.** Four lines of evidence confirmed that the method described here captured bona fide insertions present in tumour genomes. First, we searched for previously validated TAL1-proximal enhancer-associated insertions in MOLT4 and Jurkat T-ALL cells[8]; these were rediscovered by our method (Fig. 2a, Supplementary Data 1). Second, we subjected a random subset of 68 enhancer-associated insertion candidates in MOLT4 T-ALL cells to high-throughput sequencing, which confirmed that 48 (71%) of the predicted insertions were indeed present in these tumour genomes (Fig. 2b, Supplementary Table 2). Third, we carried out targeted Sanger sequencing of 37 candidate loci with insertions in MOLT4, Jurkat, Kelly, SH-SY5Y and LS174T cells, which confirmed the majority of these loci (29 of 37; 78%)

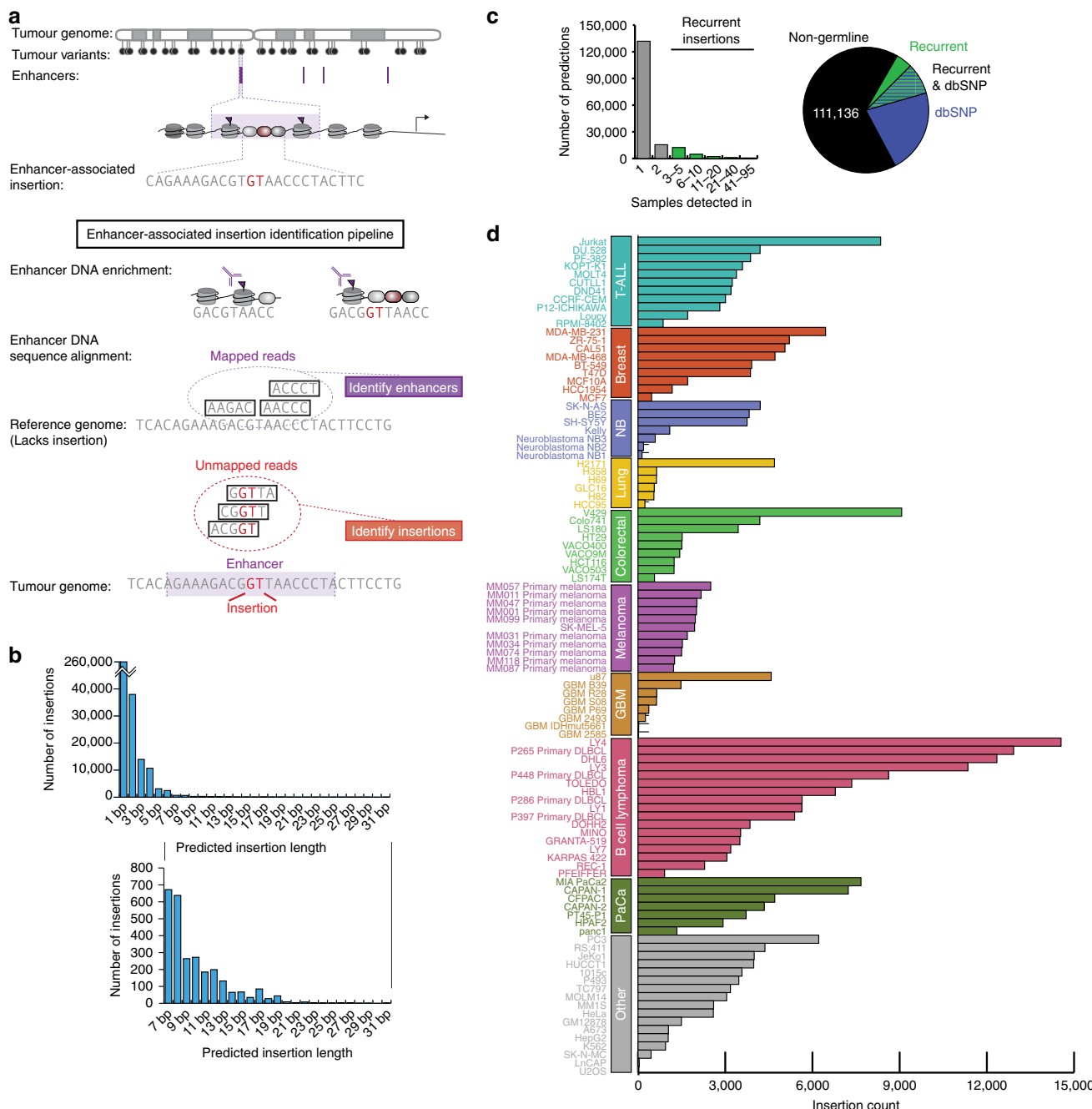

**Figure 1 | Genome-wide identification of enhancer-associated insertions. (a)** A subset of variants in tumour genomes occurs within and impacts transcriptional enhancers. ChIP-Seq experiments enrich for enhancer DNA, which may contain either reference sequences or homozygous or heterozygous variants, including insertions. Histone modifications of chromatin surround the DNA where a small insertion (red) creates a transcription factor-binding event, and this sequence is detected in the reads created in the ChIP-Seq experiment. A commonly used sequence alignment algorithm attempts to map reads to the reference genome but discards reads with insertions. Mining these initially discarded reads uncovers enhancer-associated insertions. **(b)** Genome-wide distribution of sizes of insertions predicted by our ChIP-Seq computational pipeline in 102 samples. The majority of insertions are 1 bp. **(c)** Left: Histogram showing number of samples in which an insertion is predicted. Insertions predicted in more than two samples (same location, same sequence) are considered separately because they may represent germline polymorphisms in the reference genome. Right: Pie chart depicting proportion of predicted enhancer-associated insertions present in dbSNP, predicted in many samples, or both, suggesting that these variants are acquired in the germline. **(d)** Counts of enhancer-associated insertions predicted by the pipeline processing each H3K27ac sample. Samples are grouped according to tumour type: GBM, Glioblastoma Multiforme; NB, Neuroblastoma; PaCa, Pancreatic cancer; T-ALL, T cell acute lymphoblastic leukaemia.

contained insertions (Fig. 2c, Supplementary Table 3, Supplementary Data 1). Finally, 74% (1,112 of 1,492) of predicted enhancer-associated insertions in the GM12878 cell line were supported by an Illumina Platinum Genome sequence

(Fig. 2d, Supplementary Table 4)[47]. Confirmed insertions span the size range of the predictions (1–22 bp) and include examples of homozygous and heterozygous insertions. Thirty-six of 48 high-throughput-confirmed insertions and 17 of

29 Sanger-confirmed insertions were found to be heterozygous, which is a feature we noted previously for the *TAL1*-proximal enhancer-associated insertions, although both homozygous and heterozygous insertions may alter gene expression[8]. These lines of evidence suggest that many of the predicted enhancer-associated insertions in the catalogue reflect *bona fide* insertions in tumour cell genomes. Reanalysis of whole-genome sequences from T-ALL patients showed that half of the predictions in T-ALL cell lines exist in patient genomes, demonstrating that whole-genome sequencing is capable of interrogating these variants, but current analysis pipelines commonly discard them. Only a small number of enhancer-associated insertions (25) are curated by the Catalogue of Somatic Mutations in Cancer (COSMIC) in 36 cell lines, suggesting current whole-genome analysis discards most of this class of variants[2].

**Enhancer-altering insertions associated with oncogenes**. To find the subset of insertions in our catalogue that are likely to increase enhancer activity, we filtered for insertions whose

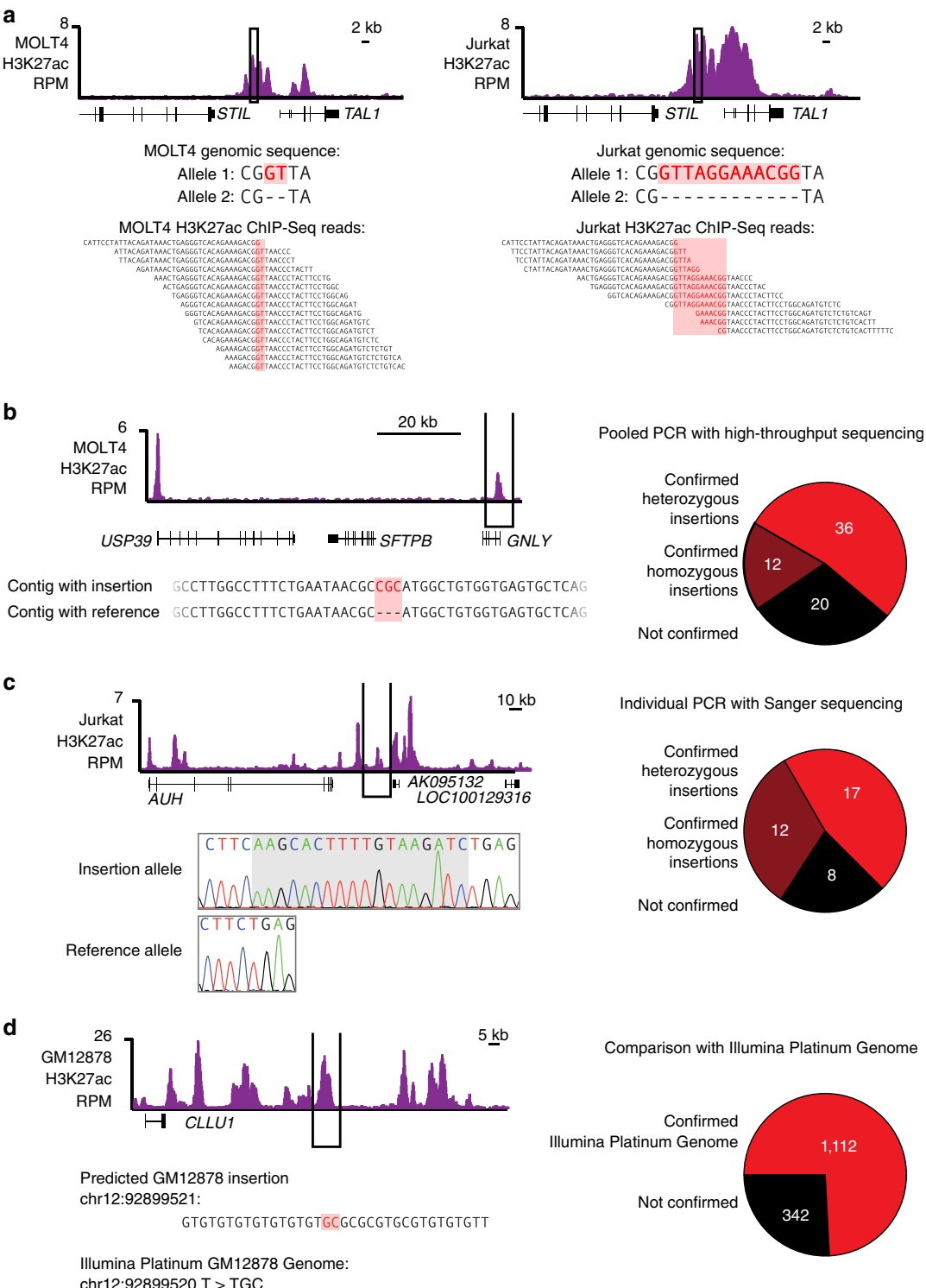

presence correlates with increased ChIP-Seq signal for nucleosomes with histone H3K27ac, which occupy active enhancers[14]. The *TAL1*-proximal insertions that cause elevated enhancer activity show such a bias[8], so we identified insertions that have increased insertion-containing reads relative to reference sequence reads. Of the 111,136 non-germline, enhancer-associated insertions, 7,213 show a twofold or greater bias in read mapping and are thus predicted to increase enhancer activity (Fig. 3a,b, Supplementary Data 1). This includes the *TAL1*-proximal insertion, which has an ∼threefold bias in coverage in Jurkat and ∼eightfold bias in coverage in MOLT4. The insertions that bias ChIP-Seq coverage by greater than twofold are more likely to be present in the genomes of tumour cells; 96% of 23 randomly selected coverage-biasing insertions were confirmed to be present as either heterozygotes or homozygotes by targeted sequencing in MOLT4 cells (Fig. 3c–e). This filtration of read-biasing insertions thus prioritizes candidate insertions that are most likely to be present and to affect enhancer activity.

To find insertions most likely to affect expression of known oncogenes, we searched for read-biasing enhancer-associated insertions that occurred within the same insulated neighbourhoods as known oncogenes (Supplementary Data 1). Insulated neighbourhoods are CCCTC-binding factor (CTCF)/cohesin-anchored DNA loops that are thought to constrain enhancer-to-gene regulatory interactions, so insertions influencing oncogene expression are likely to be inside the same insulated neighbourhoods as those oncogenes[48–50]. Many notable oncogenes occur in the same insulated neighbourhoods as enhancer-associated insertions that affect enhancer signal (Table 1). Indeed there was a significant enrichment of enhancer-associated insertions in insulated neighbourhoods that contain oncogenes ($P < 0.0001$, permutation test).

**Leukaemia oncogene targeted by enhancer-associated insertion**. We first sought to identify examples of insertion-specific enhancer activity near oncogenes in T-ALL because much is known about leukaemia oncogenes and the transcriptional control of these tumour cells[8,51,52]. A heterozygous 8-bp insertion was identified in MOLT4 T-ALL cells in the same insulated neighbourhood as *LMO2*, an established oncogenic driver in T-ALL (Fig. 4a)[53–55]. The insertion falls in a predicted SINE repeat occurrence that is uniquely mappable, so the insertion could be uniquely localized to this site using ChIP-Seq reads[56,57]. *LMO2* is not expressed in normal mature thymocytes, and its aberrant expression in these cells is thought to initiate a series of events leading to leukaemia[58], so its misregulation by enhancers is of particular interest. Interestingly, a DNase I-hypersensitive site is present at the locus in a related T-ALL cell

line, Jurkat, which does not express LMO2 (ref. 59). This is consistent with the notion that DNase I signal alone does not necessarily represent an active enhancer site, but rather the potential for enhancer formation[60]. Together, these data suggest the insertion is near a region with potential regulatory capacity near a key oncogene in leukaemia.

We next focused on the impact of the insertion on transcription using assays and analyses inspired by our initial study of a *TAL1*-proximal insertion[8]. Sanger sequencing of cloned alleles confirmed that the 8-bp insertion occurs in MOLT4 and is heterozygous (Fig. 4b). Among the T-ALL cells studied here, both the insertion and an active enhancer at this region were unique to MOLT4 cells (Fig. 4c). The location of this aberrant enhancer is not consistent with simple reactivation of a developmental enhancer (Supplementary Fig. 2A), which is a proposed general phenomenon explaining oncogenic enhancer activation[10,61]. Other, similar insertions were found in the same location in a patient-derived xenograft (TALL-12) and in 4 of 164 (2.4%) T-ALL patient samples (Fig. 4d). In three of three patients where non-tumour cells were available, we did not observe the insertion in the non-tumour cells by Sanger sequencing, suggesting that the insertion was somatically acquired in the tumour samples. In an additional cohort of T-ALL samples with whole-genome sequences from the same patient at diagnosis, remission, and relapse, we found a 45-bp insertion present at this locus at diagnosis and relapse but absent in remission DNA, suggesting this insertion was somatically acquired (Supplementary Fig. 2B).

To determine whether the 8-bp heterozygous insertion in the *LMO2* locus confers enhancer activity, the insertion allele and the reference allele were cloned into enhancer reporter vectors, and these were transfected into Jurkat T-ALL cells; the results indicate that the insertion allele has significantly more enhancer activity than the reference allele (Fig. 4e, $P < 0.001$, two-tailed Student's *t*-test). In addition, ChIP-qPCR showed that the levels of the enhancer mark H3K27ac are substantially higher on the insertion-allele than the reference allele (Fig. 4f). The insertion sequence resembles a motif recognized by MYB, which is known to interact with the H3K27ac-catalysing histone acetyltransferase CBP; ChIP-qPCR showed that MYB indeed binds preferentially to the insertion allele (Fig. 4f)[62]. MYB is part of a protein complex known to regulate the T-ALL expression programme that also includes TAL1 (ref. 8). TAL1 was also found to bind preferentially to the insertion allele (Fig. 4f), suggesting that the leukemogenic transcription-regulating complex is present at the enhancer due to the insertion.

Reanalysis of ChIP-Seq data with a modified target genome sequence showed that reads from H3K27ac, MYB and TAL1 ChIP-Seq experiments more frequently mapped to

**Figure 2 | Confirmation of predictions in the catalogue. (a)** Our computational pipeline recovered the known *TAL1*-proximal insertion in the MOLT4 and Jurkat T-ALL genomes. The insertions CG[GT]TA in MOLT4, and CG[GTTAGGAAACGG]TA noted in red, upstream of the *TAL1* gene are bound by H3K27 acetylated histones. This region was immunoprecipitated in ChIP-Seq experiments targeting acetylated H3K27, and sequence reads from this experiment contain the insertion and surrounding genomic context. **(b)** Left: Example enhancer-associated insertion in MOLT4 T-ALL cells that was confirmed by high-throughput sequencing pooled PCR products. Number of H3K27ac ChIP-Seq reads in bins at the *USP39/SFTPB/GNLY* locus is represented in purple. Annotated RefSeq genes are noted below. Representative contigs detected in the high-throughput sequencing that contain reference sequence and the predicted insertion, suggesting this insertion is heterozygous. The insertion is noted in red. Note that scaffolds were aligned to the negative strand, so insertion predicted was GCG but insertion in scaffold is GCG. Right: Pie chart summarizing numbers of predicted insertions detected using this approach. **(c)** Left: Example enhancer-associated insertion in Jurkat T-ALL cells that was confirmed by Sanger sequencing of PCR products. Number of H3K27ac ChIP-Seq reads in bins at the *AUH* locus is represented in purple. Annotated RefSeq genes are noted below. Chromatograms of Sanger sequencing of this locus are below. Chromatograms show the signal from each of four possible nucleotides at a position. Sequences of the insertions are indicated with a grey box. Right: Pie chart summarizing numbers of predicted insertions detected using this approach. **(d)** Left: Example enhancer-associated insertion in GM12878 B lymphoblastoid cells that was confirmed by the Illumina Platinum genome of these cells. Number of H3K27ac ChIP-Seq reads in bins at the *CLLU1* locus is represented in purple. The predicted insertion in genomic context is noted below in red. The Illumina-identified variant is below. Right: Pie chart summarizing numbers of predicted insertions detected using this approach.

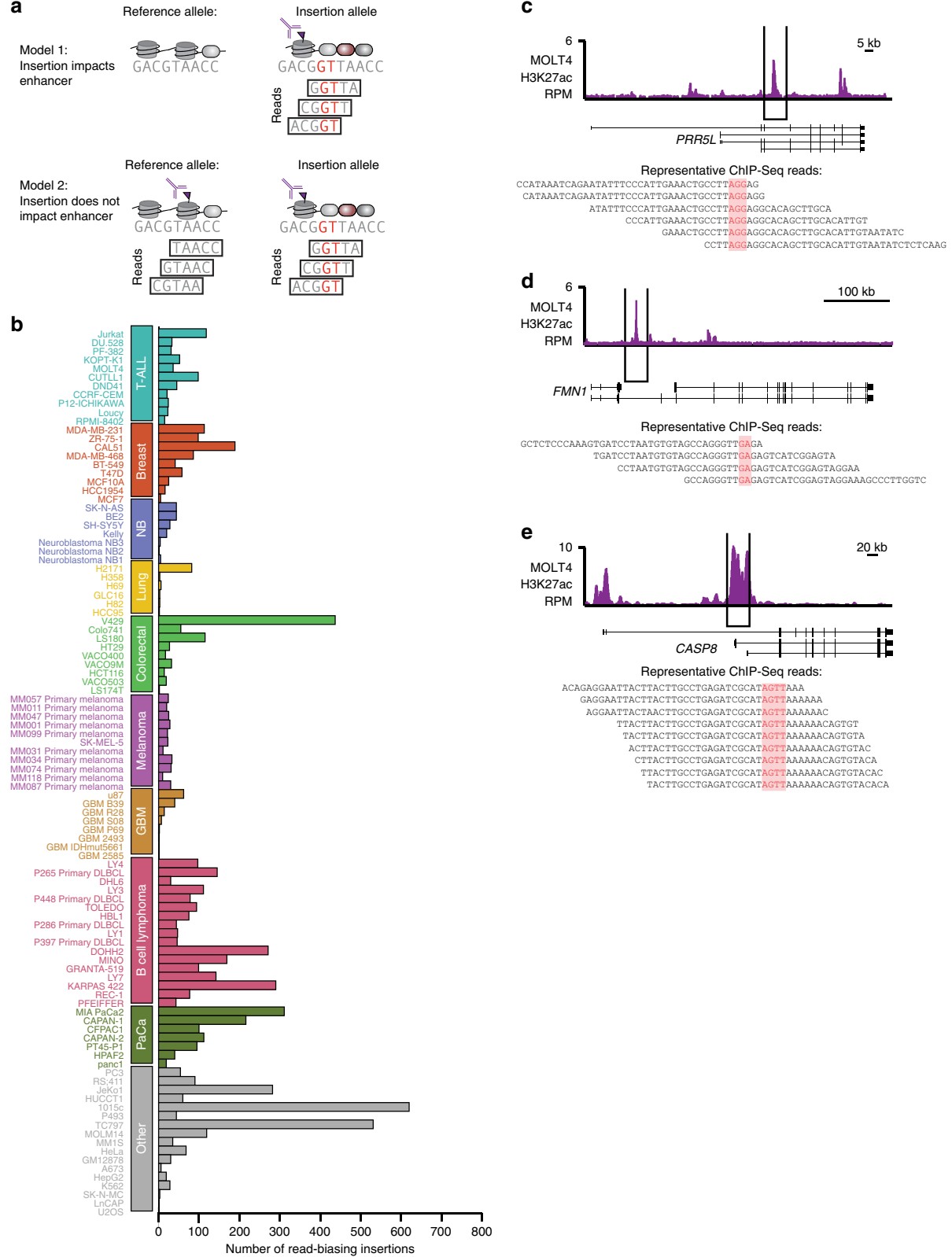

**Figure 3 | A subset of enhancer-associated insertions is predicted to alter enhancer activity.** (**a**) Cartoon depicting two plausible models of the effect of insertions on enhancers. If insertions do affect enhancers, there should be more ChIP-Seq reads for enhancer-binding proteins that contain the insertion than do not. (**b**) Counts of predicted enhancer-associated insertions in all tested samples that bias ChIP-Seq read mapping and thus are likely associated with altered enhancer activity. (**c–e**) Example enhancer-associated insertions that are predicted to alter enhancer activity. Counts of ChIP-Seq reads for H3K27ac are displayed in purple. RefSeq gene positions are noted below. ChIP-Seq reads containing the predicted insertion are noted below. The insertion is noted in red.

| Table 1 | Selected genes with enhancer-associated insertions in their insulated neighbourhoods. | |
| --- | --- |
| **Tumour type** | **Selected insertion-associated genes** |
| T-ALL | ABL, ETO2, IKZF1, IL6-RB, LMO2, MLLT1, PIM1, RUNX1, TAL1 |
| Breast | BCR, CCND1, FGFR1, KLF4, RUNX1 |
| Neuroblastoma | ETV6, FOXO1, MYCN |
| Lung | ASXL1, EML4, MYC, NF2 |
| Colorectal | ATM, AXIN2, CASP8, ERG, MSH2, PDGFB, RUNX1, TCF3 |
| Melanoma | AXIN1, BCL2, CCNE1, MYC, SS18L1, STAT3, U2AF1 |
| Glioblastoma | ERG, EXT1, SND1 |
| B cell lymphoma | ABL, BCL11A, BCL2, CASP8, CD79A, JAK2, LYL1, MSH2, PIM1, POU2AF1, RUNX1, TAL2, TNFAIP3 |
| Pancreatic | ATM |

a reference sequence containing the insertion than a reference sequence without the insertion (Fig. 4g). Using this approach, the majority of the ChIP-Seq reads include the enhancer-associated insertion sequence, supporting our interpretation. We conclude that the 8-bp heterozygous insertion in the LMO2 locus of MOLT4 cells confers elevated transcription factor binding and enhancer activity at that locus.

We next investigated whether the MOLT4 8-bp heterozygous insertion conferred heterozygous expression of LMO2. A heterozygous coding sequence variant (rs3740617) occurs in the exons of the two LMO2 alleles in MOLT4 cells, allowing us to investigate whether allele-specific LMO2 transcription takes place in these cells. Sequencing of complementary DNA (cDNA) generated from MOLT4 cells revealed that only one allele is expressed, consistent with allelic expression due to the enhancer-producing insertion (Fig. 4h). These results are consistent with the model that the 8-bp heterozygous insertion in the LMO2 locus of MOLT4 cells creates an active enhancer that drives heterozygous expression of the LMO2 oncogene. Furthermore, in one T-ALL patient with an LMO2-proximal insertion at this site, RNA sequence data were available, which showed that LMO2 was expressed from one allele (Supplementary Fig. 2C).

## Discussion

We describe here a combined experimental/computational approach that can be used for genome-wide identification of enhancer-associated variants in tumour cells. We showed that tumour cell DNA insertions identified in this way can be further studied to establish functional significance, and confirmed that an insertion at the LMO2 locus produces enhancer function in MOLT4 T-ALL cells. It is noteworthy that the target genes of this altered enhancer, and another we described previously (TAL1)[8], encode transcription factors that regulate many additional genes, so a single enhancer-inducing variant can have an outsized effect on the gene expression programme of tumour cells.

The catalogue reported here includes enhancer-associated insertions in T-ALL, breast, neuroblastoma (NB), lung, colorectal, melanoma, glioblastoma multiforme (GBM), B cell lymphoma (BCL), pancreatic and other tumour cell types (Supplementary Data 1). While this catalogue has been filtered for putatively germline variants, the catalogue may still contain both germline and somatically acquired variants. Although most cancer-driving variants described thus far are somatic in origin, and somatic variants are considered more likely to be functionally important than germline variants, the somatic or germline origin of most variants described here could not be determined for the tumour cells in this study. Nonetheless, germline variants in non-coding DNA may contribute to transcriptional misregulation of tumour oncogenes[18], so it is useful to have a catalogue of both types of enhancer-associated variants.

Cancer genome sequencing has proven valuable for the identification of numerous variants in coding DNA, but small

insertion variants in non-coding DNA that may play functional roles in tumorigenesis are less well understood. This new knowledge of enhancer-associated insertions provides a foundation for further studies to define the oncogenic contributions of this class of variants across a broad spectrum of human cancers and a new means to implicate targets for specific therapies and diagnostic approaches that empower precision medicine.

## Methods

**Cell culture.** Jurkat and MOLT4 T-ALL cells were purchased from ATCC (see Reagent validation) cultured in RPMI GlutaMAX (Invitrogen, 61,870–127), supplemented with 10% fetal bovine serum, 100 U ml$^{-1}$ penicillin and 100 µg ml$^{-1}$ streptomycin (Invitrogen, 15,140–122). CCRF-CEM, DU.528, KOPT-K1, PEER, PF-382 and P12-ICHIKAWA T-ALL cells were cultured in in RPMI 1640 (Life Technologies), supplemented with 10% fetal bovine serum, 100 U ml$^{-1}$ penicillin and 100 µg ml$^{-1}$ streptomycin (Life Technologies, 15140122). All additional cell lines were propagated according to the respective ATCC guidelines.

**ChIP-Seq.** ChIP was performed as described in Lee et al.[63] with a few adaptations. Suspension cultures were grown to a density of ~1 million cells ml$^{-1}$ before crosslinking, and adherent cell lines were crosslinked directly on the culture vessel. Crosslinking was performed for 10–15 min at room temperature by the addition of one-tenth of the volume of 11% formaldehyde solution (11% formaldehyde, 50 mM HEPES pH 7.3, 100 mM NaCl, 1 mM EDTA pH 8.0, 0.5 mM EGTA pH 8.0) to the growth media followed by five quenching with 125 mM glycine or 1 M Tris pH 7.5. Cells were washed twice with PBS, then the supernatant was aspirated and the cell pellet was flash frozen in liquid nitrogen. Frozen crosslinked cells were stored at −80 °C. Antibody-conjugated beads were prepared as follows: 100 µl of Protein G Dynabeads (Life Technologies, 10009D) were blocked with 0.5% BSA (w/v) in PBS. Magnetic beads were bound with 10 µg of anti-H3K27ac antibody (Abcam ab4729). Additional antibodies used included anti-MYB (Abcam ab45150), anti-TAL1 (Santa Cruz SC12984). These amounts were adjusted based on the number of cells used per each immunoprecipitation (see below).

Nuclei were isolated as previously described in Lee et al.[63] and sonicated in sonication buffer (50 mM HEPES-KOH pH7.5, 140 mM NaCl, 1 mM EDTA pH 8.0, 1 mM EGTA pH 8.0, 0.1% Na-Deoxycholate, 1% Triton X-100, 0.1% SDS) on a Misonix 3,000 sonicator for 10 cycles at 30 s each on ice (18–21 W) with 60 s on ice between cycles. Sonicated lysates were cleared once by centrifugation and incubated overnight at 4 °C with magnetic beads bound with antibody to enrich for DNA fragments bound by the indicated factor. 50–150 million cells were used per immunoprecipitation. For 100 million cells, 50 µl of Protein G Dynabeads and 5 µg antibody were used for each ChIP experiment, and the Dynabead and antibody amounts were scaled to cell numbers other than 100 million keeping these ratios. These ratios were used for all antibodies (H3K27ac, MYB, TAL1). After overnight incubation with the lysates, the beads were washed with sonication buffer, high-salt sonication buffer (50 mM HEPES-KOH pH 7.9, 0.5 M NaCl, 1 mM EDTA pH 8.0, 1 mM EGTA pH 8.0, 0.1% Na-Deoxycholate, 1% Triton X-100, 0.1% SDS), LiCl wash buffer (20 mM Tris-HCl pH 8.0, 250 mM LiCl, 1 mM EDTA pH 8.0, 0.5% Na-Deoxycholate, 0.5% IGEPAL C-630 0.1% SDS) and TE-0.1% Triton X-100 buffer (10 mM Tris-HCl pH 7.5, 0.1 mM EDTA pH 8.0, 0.1% Triton X-100) sequentially. DNA was eluted in elution buffer (50 mM Tris-HCL pH 8.0, 10 mM EDTA, 1% SDS). Cross-links were reversed overnight at 65 °C. RNA and protein were digested using RNase A and Proteinase K, respectively, and DNA was purified with phenol chloroform extraction and ethanol precipitation.

Purified ChIP DNA was used to prepare Illumina multiplexed sequencing libraries. Libraries for Illumina sequencing were prepared following the Illumina TruSeq DNA Sample Preparation v2 kit. Amplified libraries were size-selected using a 2% gel cassette in the Pippin Prep system from Sage Science set to capture fragments between 200 and 400 bp. Libraries were quantified by qPCR using the KAPA Biosystems Illumina Library Quantification kit according to kit protocols.

Libraries were sequenced on the Illumina HiSeq 2500 for 40 bases in single read mode.

**ChIP-Seq display.** Reads were aligned to the hg19 revision of the human reference genome using bowtie with parameters –k 2 –m 2 –sam and –l set to read length[44]. Read pileup in 50 bp bins was determined using MACS with parameters – w –S –space = 50 –shiftsize = 200 –nomodel[64]. WIG file output from MACS was visualized in the UCSC genome browser[65].

**Insertion detection pipeline.** Enhancer-associated insertions were detected using multiple alignment procedures on each H3K27ac ChIP-Seq dataset. First, to identify reads without insertions and to identify enhancers, all reads were mapped to the hg19 reference genome using bowtie with parameters –chunkmbs 256 –best –strata –m 1 –n 2 –S (ref. 44). H3K27ac reads that successfully aligned were used to locate active enhancers in each genome using MACS with parameters –p 1e-9 –keep-dup = auto[64]. Where possible, input DNA controls were used for peak calling.

Reads not initially alignable by bowtie were assembled into contigs using Edena with parameters –d 20 –c 20 and the default number of reads (5) per base to extend

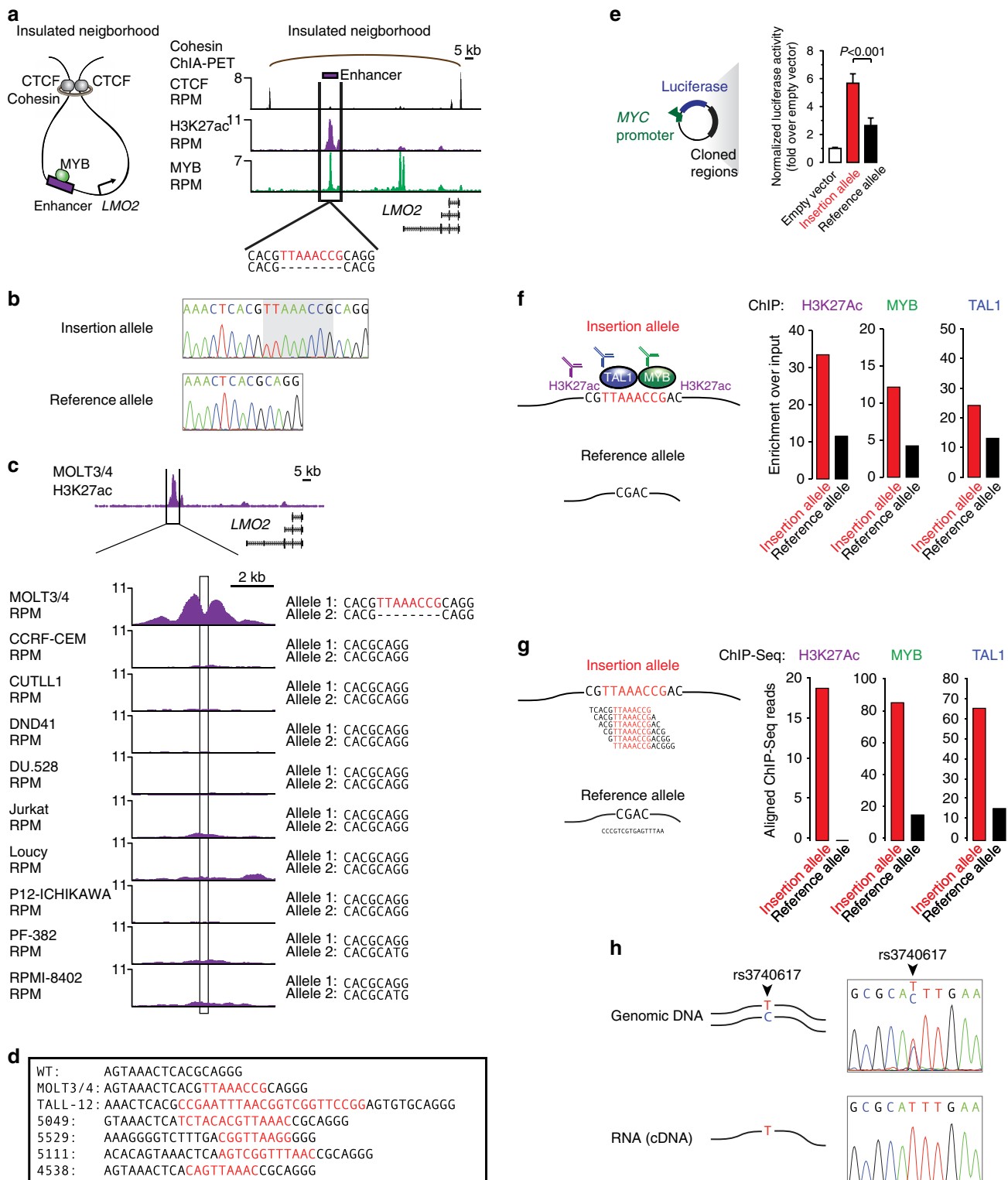

read length and increase the likelihood of finding a homologous sequence in the reference genome[66]. For samples whose read lengths were inconsistent, the first 25 bases of reads were used to build contigs. Initially unmapped reads and contigs made from these reads were aligned again to the hg19 reference genome using Bowtie 2, which permits insertions/deletions relative to the target genome, with parameters –rfg 1,1 –k 1 (ref. 11). To verify reads and contigs with insertions were robustly alignable by multiple algorithms, reads with a CIGAR string containing 'I' were used as input for BLAT with parameters –minScore = 0 –stepSize = 1. BLAT output was parsed such that each accepted read/contig hit (1) incorporated the whole read/read sequence and did not align only parts of the read/contig, (2) contained only one insertion, (3) this insertion was shorter than the read/ contig, (4) contained no BLAT-called mismatches, and the single best hit with the highest score was retained. BLAT hits were also filtered such that they had at most a 20-bp insertion, but the CIGAR string from the bowtie hit was used to determine where and what the insertion was. Bowtie 2 and BLAT hits for a read were required to be within 100 bp to be retained. Insertions in enhancers were determined by overlaps with enhancers identified above.

**Germline variation.** Enhancer-associated insertions were analysed for (1) presence in dbSNP[46] and (2) recurrence across samples. The sequences and hg19 positions of dbSNP 144 were downloaded from the UCSC table browser and converted into coordinate + allele identifiers. Predicted variants with the same position and sequence as the dbSNP identifiers were considered present in dbSNP and thus germline. In the absence of matched germline data for the cell lines, we reasoned that recurrent insertions across samples from many, unrelated individuals likely represent germline variation. Any insertion with the same position and sequence identified in more than two samples was considered germline, because (1) the TAL1-proximal insertion existed in only two of our cell lines and (2) the majority of these insertions was present in dbSNP.

**Deletions.** Deletions were determined using only contigs. Contigs were aligned using bowtie 2 as described above and reads with CIGAR strings containing 'D' were used as input for BLAT as described above. BLAT alignments were processed as aligned above but CIGAR strings from bowtie2 were used for downstream analysis. Hits with a CIGAR string containing one D, two Ms and zero Is were considered deletions.

**Assignment of predictions to genes.** Insertions were assigned to genes considered active that are in the same insulated neighbourhood as the insertion. DNA interactions where cohesin is present were previously defined in Jurkat cells using SMC1 Chromatin Interaction Analysis by Paired-end Tag sequencing (ChIA-PET)[50]. We filtered these loops for high-confidence interactions that have an FDR < 0.2 and have CTCF-enriched regions contacting both loop anchors. CTCF peaks were defined using MACS with input control and parameters –keep-dup = 1 and –p 1e-9. Because these loops can be nested, one insertion can be contained within multiple loops, so the smallest loop containing an insertion was considered. Insertions were associated with RefSeq genes whose TSS was also in the same insulated neighbourhood. Oncogenes are defined as all genes listed in the COSMIC cancer gene census[2].

**Confirming predictions by high-throughput sequencing.** A portion of the predicted indels was confirmed using an Illumina MiSeq. First, the genomic region in question was amplified with PCR. PCR primers were designed ∼100–150 bp upstream and downstream of the predicted indel using Primer3 (total amplicon size ranged from 200 to 300 bp)[67]. Primer sequences are available in Supplementary Table 2. PCR was carried out with Phusion Flash High Fidelity PCR Mix (Fischer Scientific F-548S) using standard conditions and

genomic DNA from the cell line in which the indel was predicted. The products were pooled and purified. The pooled PCRs were then sequenced on a MiSeq with 150 × 150 paired-end reads. The raw output is available in dbGaP under accession phs001242.v1.p1.

The presence or absence of the indel was determined through an alignment strategy. Small custom genomes were created for the reference- and insertion-containing sequences at each predicted insertion with 250 nucleotides upstream and downstream of the insertion position. Reads were assembled into contigs using Edena[66] with parameters –d 20 –c 20 -minCoverage 5 to get a minimum of 5x coverage of ease nucleotide in the contig. Contigs were mapped to the small custom genomes using bowtie with parameters –chunkmbs 256 –best –strata –f –m 1 –n 0 –p 10 to allow for zero mismatches between the small custom genome and each alignable contig. Alleles were considered present if there were any contigs that aligned and contacted the position of the predicted insertion.

**Confirming predictions with Sanger sequencing.** A portion of the predicted indels was confirmed using Sanger sequencing. First the genomic region in question was amplified with PCR. PCR primers were designed ∼250 bp upstream and downstream of the predicted indel using Primer3 (ref. 67). Primer sequences are available in Supplementary Table 3. PCR was carried out with Phusion Flash High Fidelity PCR Mix (Fischer Scientific F-548S) using standard conditions and genomic DNA from the cell line in which the indel was predicted in. The products were purified (Qiagen 28104) and Sanger sequenced in the forward and reverse direction using the original primers. The presence or absence of the indel was called through examination of the chromatogram and comparison with the reference genome. The sequence of the insertion was determined either by manual deconvolution of the chromatogram or cloning. When cloning was used, PCR products were cloned into the pGL3 vector and at least six individual clones were sequenced.

**Patient-derived xenografts/patient samples.** Diagnostic DNA samples were available from 164 paediatric and young adult T-ALL patients (age 1–25) entered into the UKALL2003 trial, excluding those with bi-phenotypic leukaemia or T-cell lymphoma. Ethical approval for the trial was obtained from the Scottish Multi-centre Research Ethics Committee and informed consent was obtained in accordance with the Declaration of Helsinki. The trial is registered at http://www.controlled-trials.com under ISRCTN number 07355119. Mutation screening for recurrence of the LMO2 enhancer mutation was performed using denaturing high-performance liquid chromatography and Sanger sequencing.

**Luciferase enhancer-reporter assays.** Luciferase reporter assays were performed as previously described[68] with modifications. The candidate enhancer region (∼600 bp) around the LMO2 locus was cloned into a pGL3 (Promega) reporter vector (BamHI-SalI sites) that contains a Firefly luciferase gene driven by a minimal c-MYC promoter[69]. The candidate enhancer region around LMO2 was PCR-amplified using the following primer sequences (5′–3′): ACTTTGCC TTTCCCCAGTTGC and ATGGCCTTTCTGAGCCTTCC. MOLT4 genomic DNA was used as template DNA in the PCR reactions. The sequences of the cloned candidate enhancer containing the LMO2-proximal or the reference sequence were verified by Sanger sequencing. $2 \times 10^5$ Jurkat T-ALL cells were transfected with 475 ng of the reporters using MOLT4 Avalanche transfection reagent (EZ Biosystems). 25 ng of a Renilla luciferase control plasmid (pRL-SV40; Promega) was co-transfected as a normalization control. After 40 h of incubation luciferase activity was measured using the Dual-Luciferase Reporter Assay System (Promega). All luciferase reporter assays were performed in quadruplicates. Luciferase activity was normalized to the activity measured in cells transfected with a construct containing only the promoter.

**Figure 4 | A confirmed insertion alters the regulation of a T-ALL oncogene.** (**a**) Insertion near LMO2 in MOLT4 T cell acute lymphoblastic leukaemia cells. (Left) representation of an insulated neighbourhood, which is a loop between distal CTCF- and cohesin-bound sites. The MYB-bound LMO2 enhancer and LMO2 gene are within the neighbourhood. (Right) An insulated neighbourhood defined in Jurkat T-ALL cells connecting CTCF-bound sites encompasses LMO2 and its enhancer. Tracks of H3K27ac and MYB ChIP-Seq signal at the LMO2 locus with predicted insertion in the MOLT4 genome and protein-coding oncogene below. Region containing the insertion is indicated in black. Inserted sequence is in red. Scale bar represents 5,000 bases. (**b**) Sanger sequencing chromatograms of MOLT4 alleles separately cloned from the heterozygous insertion in the LMO2 enhancer. (**c**) ChIP-Seq signal at the LMO2 enhancer across 10 T-ALL samples. The sequence at the LMO2 enhancer-associated insertion is noted. The KOPT-K1 genome contains a translocation near LMO2 and was not included in the display. Scale bar represents 2,000 bases. (**d**) Patient genomes contain insertions at the LMO2 enhancer locus, noted in red. (**e**) Enhancer activity of the luciferase reporter is significantly higher for the region containing the insertion allele compared to the region not containing the insertion allele (P < 0.001, two-tailed Student's t-test). The mean is plotted, and error bars indicate s.d. from four replicates. (**f**) Allele-specific ChIP-qPCR bar charts showing quantitative H3K27ac, TAL1 and MYB binding at the region containing the insertion; the allele with the insertion is preferentially bound by all three. ChIP-qPCR was performed for each of the three factors with primers that include or exclude the insertion. Enrichment over input DNA (ΔΔcT) is plotted. (**g**) ChIP-Seq reads for H3K27ac, TAL1, and MYB preferentially aligned to reference sequences containing the LMO2-proximal insertion. Counts of reads aligning to an insertion-including reference (red) and insertion-excluding reference (black) are displayed as barplots. (**h**) Sanger sequencing chromatograms of gDNA and cDNA show that the LMO2 gene is expressed from one allele in MOLT4 cells. (top) A coding SNP in LMO2 is confirmed to be heterozygous by sequencing genomic DNA in an LMO2 exon (gDNA). (bottom) Sanger sequencing of cDNA reverse-transcribed from mRNA shows only one heterozygous LMO2 allele is transcribed.

**Allele-specific ChIP-qPCR.** ChIP was performed as described in the ChIP-Seq section above with antibody dilutions as described. Additional antibodies used included anti-MYB (Abcam ab45150) and anti-TAL1 (Santa Cruz sc12984). Allele-specific enrichment was detected using allele-specific primers in quantitative real-time PCR performed on a 7000 AB Detection System according to the manufacturer's instructions (Applied Biosystems). The following primers were used:

Insertion containing allele: (5′–3′) TCCTGCCCTGCGGTTTAACG, and (5′–3′) GATCTGCTTCTTGGAGAGCTGC

Reference allele: (5′–3′) GCCCTGCGTGAGTTTACTGTG and (5′–3′) GATCTGCTTCTTGGAGAGCTGC

Enrichment was calculated using the delta Ct method against a negative control region amplified by the following primers: (5′–3′) CCCACCTTGTGTTCA AATGCTGA and (5′–3′) ACGCTTTTCTTCTGCCTTCTGC. The values calculated in the ChIP samples were subsequently normalized against those measured using the ChIP input DNA as a template in the qPCR reaction.

**Allele-specific ChIP-seq coverage.** If an enhancer-associated insertion alters enhancer activity, it is predicted to alter the number of reads coming from each allele in ChIP-seq experiments. For each predicted insertion, two small genomes were created containing only the reference sequence or the reference sequence modified with the predicted insertion. Small genomes were two times the read length, centred on the insertion locus, and the reference sequence was taken from hg19.

For expediency, bowtie (parameters –chunkmbs 256 –best –strata –n 2) was used to align to a file containing all reference or all insertion sequences to create a list of all possible mappable reads in this strategy. The –m parameter that determines the maximum number of allowed mappings was set to the number of insertions predicted in the sample to allow reads that may map to multiple insertion sites.

The files containing all reference-mapping or insertion-mapping reads was then used to align to each reference small genome or insertion small genome for each insertion. This was accomplished with bowtie with parameters –chunkmbs 256 –best –strata –m 1 –n 2. Reads mapping to each allele were counted.

**Allele-specific RNA assay.** To detect allele-specific *LMO2* expression, we exploited the heterozygosity at the rs3740617 SNP in MOLT4 cells. This SNP is located in the 3′untranslated regions of the LMO2 transcript. The region containing the SNP was amplified by PCR using the following primers: (5′–3′) GTCCTTCTGTCACCTTGAAGTG and (5′–3′) TATGCCAGATCCAAA TGCCAG. Either genomic DNA of MOLT4 cells or cDNA was used as a template in the PCR reaction. To generate cDNA, RNA was isolated using the RNeasy Plus purification kit (Promega), and reverse transcribed using oligo-dT primers and SuperScript III reverse transcriptase (Life Technologies 18080044) according to the manufacturers' instructions. The fragments were gel-purified (Qiagen Gel Extraction Kit) and submitted to Sanger sequencing using the primers used for amplification. The distribution of the alleles containing the two nucleotide variant at the SNP position in the genomic DNA and cDNA was estimated by inspection of the Chromatograms of the Sanger sequencing reactions.

**Processing insertions in COSMIC.** To identify predicted enhancer-associated insertions in patient data, we downloaded grch37 coordinates and sequences from COSMIC v75. These were parsed into coordinate + sequence identifiers. The collapsed union of all enhancers from all samples in this study was created and used to find COSMIC insertions in enhancers. A strict overlap was required to count the number of patient insertions that could exist in the enhancers of our cell lines.

Thirty-six of our 82 cell lines were analysed by COSMIC's Cell Line Project: A673, CAPAN-1, Capan-2, CCRF-CEM, CFPAC-1, COLO-741, DND-41, DOHH-2, GRANTA-519, NCI-H2171, NCI-H358, NCI-H69, NCI-H82, HCC1954, HCT-116, HeLa, JEKO-1, Jurkat, KARPAS-422, K-562, KELLY, LOUCY, MCF7, MDA-MB-231, MDA-MB-468, MIA-PaCa-2, MM1S, MOLT-4, OCI-LY7, P12-ICHIKAWA, PC-3, RPMI-8402, RS4-11, SK-MEL-5, SK-N-AS, T47D. The COSMIC enhancer-associated insertions from each sample were compared against the 118,514 predicted enhancer-associated insertions in these lines using position + sequence.

**Analysis of patient genomes.** We analysed 60 tumour-normal matched samples from Shanghai Children's Medical Center (SCMC, N = 33) and St Jude Children's Research Hospital (SJCRH, N = 27) collected as part of an ongoing collaboration between both, including 43 diagnosis tumours and 17 relapsed samples. Samples were collected and analysed by whole genome sequencing (tumour and normal) and RNA sequencing (tumour only) at SCMC or SJCRH accordingly. We compared the variation between the tumour and normal samples to verify if any candidate insertion from our analysis was somatically acquired. Insertions proximal to *TAL1* and *LMO2* were discovered to be somatically acquired in this cohort. RNA-seq data were analysed to identify allele-specific production of reads consistent with heterozygous expression from this locus. Eleven SNPs were

confirmed to be heterozygous in the *LMO2* locus. Allele-specific pileup of reads from whole-genome sequencing and RNA-seq data were calculated.

**Reagent validation.** ab4729, which is used for all newly published ChIP-seq, ab45150, which is used for MYB ChIP-PCR, and sc-12984, which is used for TAL1 ChIP-PCR are human ChIP-grade antibodies: http://1degreebio.org/reagents/product/101583/?qid=1356150

http://1degreebio.org/reagents/product/101339/?qid=1386512

https://www.citeab.com/antibodies/829388-sc-12984-tal1-c-21/

In addition, sc-12984 has been validated in our lab to specifically bind TAL1 (ref. 51).

RPMI-8402 is listed as a misidentified cell line where authentic stock is known to exist[70]. For our previous publication, this stock was purchased from DSMZ, which is the recommended source of authentic stock (GSM1442003).

For this study, we produced H3K27ac ChIP-Seq datasets in eight cell lines: CCRF-CEM, DU.528, KOPT-K1, Loucy, MOLT-4, P12-ICHIKAWA, PF-382, SK-N-AS.

CCRF-CEM, MOLT4, Jurkat, Loucy and SK-N-AS cells were purchased from ATCC with catalogue numbers CCL-119, CRL-1582, TIB-152, CRL-2629 and CRL-2137, respectively.

DU.528 cells were acquired from J. Kurtzberg at Duke.

KOPT-K1 cells were acquired from S. Nakazawa at Yamanashi Medical University in Japan.

PF-382 cells were purchased from DSMZ with catalogue number AAC 38.

Cell lines were all authenticated by DNA fingerprinted with small tandem repeat profiling by Genomics Core Services in Molecular Biology Core Facilities, DFCI. CCRF-CEM, MOLT4, Jurkat, Loucy, DU.528, KOPT-K1 and PF-382 cells were fingerprinted in February 2016. SK-N-AS cells were fingerprinted in March 2015. Cell lines were all tested for mycoplasma contamination using MycoAlert, and all tested negative before sample preparation.

**Data availability.** The ChIP-Seq and control datasets generated for this study are available in the GEO repository https://www.ncbi.nlm.nih.gov/geo/query/acc.cgi?acc=GSE76783. The ChIP-Seq datasets analysed in the current study are noted in Supplementary Table 1.

The hg19 genomic locations of germline variants in dbSNP (All SNPs v 144) are available via the UCSC genome browser:

https://genome.ucsc.edu/cgi-bin/hg

The identity and hg19 locations of mutations in cell line genomes are available on request from COSMIC: http://cancer.sanger.ac.uk/cell_lines

The identity and hg19 locations of mutations in patient genomes are available on request from COSMIC: http://cancer.sanger.ac.uk/wgs

The cohesin (SMC1) interactions used to define insulated neighbourhoods for gene assignment are available in a previous publication[50] as Table S2A. https://www.ncbi.nlm.nih.gov/pmc/articles/PMC4884612/bin/NIHMS783783-supplement-Table_S2.pdf

Variants in the GM12878 Illumina Platinum Genome are available from Illumina: ftp://platgene_ro@ussd-ftp.illumina.com/older_releases/hg19/8.0.1/NA12878/NA12878.vcf.gz

The hg19 genomic locations of repeat elements from RepeatMasker used to qualify the *LMO2*-proximal enhancer region as a low-fidelity SINE repeat are available from the UCSC genome browser: https://genome.ucsc.edu/cgi-bin/hg

The high-throughput sequencing results have been deposited on the GEO database with accession number GSE76783.

Data from high-throughput sequencing of predicted insertions in the MOLT4 cell line have been deposted in dbGaP with accession number phs001242.v1.p1. The remaining data are available within the Article file and Supplementary Information or available from the author upon request.

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

## Acknowledgements

We thank David Weinstock for providing patient-derived xenograft data, Lars Anders and Heather Hoke for ChIP-Seq data, Andrea Califano for experimental reagents, and Dave Orlando for helpful discussion. Primary childhood leukaemia samples used in this

study were provided by the Bloodwise Childhood Leukaemia Cell Bank working with the laboratory teams in the Bristol Genetics Laboratory, Southmead Hospital, Bristol: Molecular Biology Laboratory, Royal Hospital for Sick Children, Glasgow: Molecular Haematology Laboratory, Royal London Hospital, London: Molecular Genetics Service and Sheffield Children's Hospital, Sheffield. B.J.A. is the Hope Funds for Cancer Research Grillo-Marxuach Family Fellow. M.R.M. is a Bloodwise Bennett Fellow, and funded by grants from CRUK, Gabrielle's Angels Foundation, and the Freemasons' Grand Charity. S.R. is supported by Bloodwise. Z.L. was supported by Alex's Lemonade Stand Foundation's Young Research Investigator grant. D.H. is supported by an Erwin Schrodinger Fellowship (J3490) from the Austrian Science Fund (FWF). A.S.W. is supported by Ludwig Graduate Fellowship funds. This work was supported by the National Institutes of Health grants HG002668 and CA109901 (R.A.Y.). This work was supported in part by the National Cancer Institute Grant R35 CA210064 (A.T.L.). This work was supported in part by the American Lebanese Syrian Associated Charities of St Jude Children's Research Hospital, by National Cancer Institute Grants P30 CA021765 (St Jude Cancer Center Support Grant), and by National Institute of General Medical Sciences Grant P50 GM115279 (to J.Z.). Partly supported by grants from the National Natural Science Foundation of China (81670174 to B.L.), the Science and Technology Commission of Pudong New Area Foundation (PKJ2014-Y02 to B.L.), and Viva China Children's Cancer Foundation. R.A.Y. is a founder of Syros Pharmaceuticals and Marauder Therapeutics.

## Author contributions

B.J.A., A.T.L., M.R.M. and R.A.Y. conceived the study. B.J.A. designed the pipeline, constructed the catalogue and wrote the paper with input from all authors. D.H. performed allele-specific ChIP, luciferase assays and expression analysis. D.H., A.S.W., N.K. and C.H.L. performed confirmation of predictions in cell lines. S.R. performed Sanger sequencing of predictions in primary patient samples. Y.L., B.L., S.S. and J.Z., analysed T-ALL patient genomes for mutations. D.H., N.K., Z.L., N.W-L., A.L. and J.E. performed ChIP-Seq experiments. A.T.L., M.R.M. and R.A.Y. supervised the project.

## Additional information

**Competing financial interests:** The authors declare no competing financial interests.

**Publisher's note**: 

DOI: 10.1038/ncomms15797   OPEN

# Corrigendum: Small genomic insertions form enhancers that misregulate oncogenes

Brian J. Abraham, Denes Hnisz, Abraham S. Weintraub, Nicholas Kwiatkowski, Charles H. Li, Zhaodong Li, Nina Weichert-Leahey, Sunniyat Rahman, Yu Liu, Julia Etchin, Benshang Li, Shuhong Shen, Tong Ihn Lee, Jinghui Zhang, A. Thomas Look, Marc R. Mansour & Richard A. Young

*Nature Communications* 8:14385 doi: 10.1038/ncomms14385 (2017); Published 9 Feb 2017; Updated 1 Jun 2017

In the original version of Supplementary Data 1 associated with this Article, the list of predicted enhancer-associated insertions was inadvertently truncated. The HTML has now been updated to include the correct version of the Supplementary Data 1.

