## [Peer Review File · Nature Communications]

Reviewers' comments:

Reviewer #1 (Remarks to the Author):

The manuscript by Abraham and colaboradores describe an original approach to detect insertions in enhancers. It consist in using the ChIP-seq data of H3K27ac (a enhancer-associated histone mark) to detect insertions in cancer cell lines and tumor samples. The idea is original and the analysis is straightforward. The authors follow up an insertion in one candidate oncogenic enhancer regulating LMO2 gene in T-ALL cells, confirming that it leads to an active enhancer and overexpression of LMO2 gene.

The main concern I have is the number of false positives that may be found with this approach, and although the authors use the approach as a discovery step to then filter and focus on one or few candidates, the manuscript will benefit from a more accurate estimation or discussion of the rate of false positive in each step.

- The authors describe 101,946 insertions and 33,703 additional ones that were discarded as they are shared by multiple independent tumor samples and are thus regarded as possible germline variation. Are the authors surprised by the large number of insertions detected?

- The authors say: "The pipeline was adapted to predict deletions in a similar manner, but we chose to focus on insertions for further study (Supplementary Data 3)." Why they choose to focus in insertions only? Would be useful a brief description of the rationale behind this decision.

- 22 out of 33 candidate loci with insertions were validated. How the 33 candidate loci with insertions were selected? Which criteria was used? Is this proportion of 22 over 33 a representation of the true positive rate overall? This is only true if the 33 were picked randomly to represent the variation in 101,946 insertions detected.

- Even among the real insertions detected there is still an important issue to solve, if they are somatic or germline variation. A number of detected insertions (33,703) were discarded as they appeared in more than one tumor sample, however an unknown proportion of the ones that appear in only one sample may also be germline variation. Could authors comment on this?

- Also, among the 33,703 discarded insertions there may be recurrent oncogenic somatic insertions, although most I agree are probable germline variation. Based on the results from a previous article by the authors (Mansour et al., Science 2014), a recurrent insertion in the enhancer of TAL1 (exactly the same insertion was found in 3 patients) was shown to be oncogenic. Authors should also discuss this possibility.

- An unknown number of the detected insertions are real and somatic. Among those probably only very few are oncogenic, as most of those are likely to be passenger insertions. To find the probable oncogenic insertions in enhancers the authors focus on those that occurred within the same insulated neighborhoods as known oncogenes. The rest of the manuscript focus in the detailed analysis of an insertion in an enhancer of LMO2 gene in T-ALL cells.

Reviewer #2 (Remarks to the Author):

Brian Abraham et al report a strategy designed to identify insertions in non-coding regions of the genome that may contribute to aberrant expression of oncogenes. The manuscript contains what appears to be an intuitive computational pipeline for identifying such insertions, and then also some validation of a regulatory element in the LMO2 gene locus. While the data contained in this manuscript seem to be robust, there are several instances where in my view the study is rather

preliminary. Consequently there is little certainty for any potential for wider future impact with the paper as it stands.

Major Points:

1) The actual nature of these enhancer insertions remains unclear. It was impressive but possibly somewhat underhand that the authors managed to write this paper in a way that suggests the mutations are somatically acquired without ever stating it. And the reason they are not stating it is because they have no evidence for it. But that leaves the question wide open as to what proportion of the so-called insertions are actually polymorphisms, and which ones are acquired mutations that cause ectopic expression of oncogenes. Without any data on this critical point, the paper is inconclusive and very preliminary. Absolutely critical experiments are:

i) Analysis of constitutional/germline DNA from matched tumor/normal samples. I realize this may be difficult for the cell lines, but then they should look at patient samples where normal control DNA would often be available (indeed later in the paper there is talk of using some patient samples)

ii) A thorough investigation of how many of the so-called insertions are polymorphisms. Detailed analysis of for example the 10,000 genome project data needs to be undertaken, as well as other publicly available full genome sequence data collections. It will be possible to screen bioinformatically through the raw data of these collections specifically with the "insertion region sequences" and then see whether or not there are any matches in normal individuals.

2) It was unclear to me why some of the so-called insertions were homozygous. For a "gain of function" event such as the one they imply for LMO2, alteration of a single allele should be enough. Therefore, what do the authors think is going on in the homozygous samples? Is it loss of function leading to loss of expression of a tumour suppressor? But then why would this be picked up with acetylated histones? Or is it acquired homozygosity through mitotic recombination? Or is it indeed a polymorphism that happens to be homozygous in this particular sample (see point above in relation to what proportion of the events are actually polymorphisms).

3) What is the background mutation rate in these cultured cancer cell lines? Cancer cells commonly have defects in DNA repair, and cancer cell lines often have very messed-up genomes. How can the authors exclude that such in vitro processes are not major contributors to what they describe?

4) It was difficult to assess which Chip-seq experiments were newly done for this paper, and which were old (bottom of page 3). Why not just be upfront and give the actual numbers?

5) A quick scan of the literature suggests that a fair amount is known about the regulation of LMO2. Does the region picked up in this paper correspond to any of the previously mapped LMO2 regulatory regions? Is it therefore perhaps a known element with added ectopic activity, or are the authors suggesting generation of a completely new element de novo?

6) Table 1: Analysis needs to be performed to determine whether the number of oncogenes observed is more than one would expect by chance. If not, the table may be meaningless.

Thank you for the reviews and comments on our manuscript “Oncogenic Enhancers from Small Genomic Insertions.” The reviewers noted that we describe an original approach to detect insertions in enhancers, provide an intuitive computational pipeline for identifying such insertions and have data that seems to be robust. They also noted concerns, which include a need to provide greater clarity with respect to success rates, the rationale behind selecting insertions for further study, and whether variants are somatic or germline in origin. We have addressed these concerns in our revised manuscript by conducting additional computational and experimental analyses and by clarifying the text. These revisions include the results of testing 79 additional predicted insertions, integrating our predicted insertions with existing polymorphism data, and testing normal and tumor cells from the same patient. In addition, we have modified the samples investigated in our study to conform to the Nature editorial policy on reporting life sciences research. We thank the reviewers for their guidance and believe this revised manuscript is considerably improved.

A detailed response to the comments is included below.

Reviewer #1 (Remarks to the Author):

The manuscript by Abraham and collaborators describes an original approach to detect insertions in enhancers. It consists in using the ChIP-seq data of H3K27ac (a enhancer-associated histone mark) to detect insertions in cancer cell lines and tumor samples. The idea is original and the analysis is straightforward. The authors follow up an insertion in one candidate oncogenic enhancer regulating LMO2 gene in T-ALL cells, confirming that it leads to an active enhancer and overexpression of LMO2 gene.

The main concern I have is the number of false positives that may be found with this approach, and although the authors use the approach as a discovery step to then filter and focus on one or few candidates, the manuscript will benefit from a more accurate estimation or discussion of the rate of false positive in each step.

We now provide an improved description of the pipeline and a discussion of error rates in the manuscript, including a discussion of the false positives at various steps in the pipeline. Based on targeted sequencing of a randomly selected subset of predicted insertions, the overall percentage of false-positive insertion predictions appears to be approximately 27% (28/105 absent), but the subset of insertions predicted to affect enhancer activity appears to have a smaller false-positive percentage of 4% (1/23 absent). We have added the following description to the text on page 4:

Four lines of evidence confirmed that the method described here captured bona fide insertions present in tumour genomes. First, we searched for

previously validated TAL1-proximal enhancer-associated insertions in MOLT4 and Jurkat T-ALL cells⁸; these were rediscovered in our catalogue (Figure 2A). Second, we subjected a random subset of 68 enhancer-associated insertion candidates in MOLT4 T-ALL cells to high-throughput sequencing, which confirmed that 48 (71%) of the predicted insertions were indeed present in these tumour genomes (Figure 2B, Supplemental Table 2). Third, we carried out targeted Sanger sequencing of 37 candidate loci with insertions in MOLT4, Jurkat, Kelly, SH-SY5Y and LS174T cells, which confirmed the majority of these loci (29 of 37; 78%) contained insertions (Figure 2C, Supplemental Table 3, Supplementary Data 1). Finally, 74% (1,112 of 1,492) predicted enhancer-associated insertions in the GM12878 cell line were supported by an Illumina Platinum Genome sequence (Figure 2D, Supplemental Table 3)³⁸. Confirmed insertions span the size range of the predictions (1-22 bp) and include examples of homozygous and heterozygous insertions. Thirty-six of 48 high-throughput-confirmed insertions and 17 of 29 Sanger-confirmed insertions were found to be heterozygous, which is a feature we noted previously for the TAL1-proximal enhancer-associated insertions, although both homozygous and heterozygous insertions may alter gene expression⁸. These lines of evidence suggest that many of the predicted enhancer-associated insertions in the catalogue reflect bona fide insertions in tumour cell genomes.

- The authors describe 101,946 insertions and 33,703 additional ones that were discarded as they are shared by multiple independent tumor samples and are thus regarded as possible germline variation. Are the authors surprised by the large number of insertions detected?

The number of enhancer-associated insertions detected by our pipeline is not surprising given the baseline rate of indels per genome reported by others. Based on studies of the diploid genomes of individuals, it has been estimated that 275,512 homozygous insertions and 45,622 heterozygous indels exist in an individual's genome relative to the reference genome [2]. We estimate that enhancers encompass ~2% of the genome in any one cell type, so we might expect ~6,000 insertions per genome relative to the NCBI reference. We find an average of 3,200 insertions in the enhancers of each genome. This number is within expectations for baseline genome variation. Furthermore, only a subset of these insertions is predicted to alter the activity of any of these enhancers. We have now tested all predicted insertions for their ability to bias ChIP-Seq reads, which serves as a proxy for actual protein binding. There are substantially fewer enhancer-associated insertions (~7,200) across all samples that, by our metrics, are predicted to affect enhancer activity. While the insertions we predict are generally present in enhancers, comparatively few are predicted to affect enhancer activity. We have added a discussion of this to the text on page 5:

To find the subset of insertions in our catalogue that are likely to increase enhancer activity, we filtered for insertions whose presence correlates with increased ChIP-Seq signal for nucleosomes with histone H3K27ac, which occupy active enhancers¹⁴. The TAL1-proximal insertions that cause elevated enhancer activity show such a bias⁸, so we identified insertions that have increased insertion-containing reads relative to reference sequence reads. 7,213 of 111,136 non-germline, enhancer-associated insertions show a two-fold or greater bias in read mapping and are thus predicted to increase enhancer activity (Figure 3A, B, Supplementary Data 1). This includes the TAL1-proximal insertion, which has an ~3-fold bias in coverage in Jurkat and ~8-fold bias in coverage in MOLT4. The insertions that bias ChIP-Seq coverage by greater than 2-fold are more likely to be present in the genomes of tumour cells; 96% of 23 randomly selected coverage-biasing insertions were confirmed to be present as either heterozygotes or homozygotes by targeted sequencing in MOLT4 cells (Figure 3C, 3D, 3E). This filtration of read-biasing insertions thus prioritizes candidate insertions that are most likely to be present and to affect enhancer activity.

- The authors say: "The pipeline was adapted to predict deletions in a similar manner, but we chose to focus on insertions for further study (Supplementary Data 3)." Why they choose to focus in insertions only? Would be useful a brief description of the rationale behind this decision.

Our motivation for focusing on insertions comes largely from our previous experience examining one small insertion in T-ALLs whereby a small insertion alters enhancer activity near an oncogene [1]. We believe this represents the simplest model that a small insertion nucleates an enhancer that drives expression of oncogenes, so we set out to determine how frequently this occurs across other tumor genomes. It is indeed likely that small deletions contribute in a similar manner to oncogene misregulation. For this reason, we have included a set of predicted deletions that may be of interest to the community. We have mentioned this in the text on page 4:

The pipeline was adapted to predict deletions in a similar manner (Supplementary Data 2), but we chose to focus on insertions for further study because of our previous experience.

- 22 out of 33 candidate loci with insertions were validated. How the 33 candidate loci with insertions were selected? Which criteria was used? Is this proportion of 22 over 33 a representation of the true positive rate overall? This is only true if the 33 were picked randomly to represent the variation in 101,946 insertions detected.

In the original submitted manuscript, we hand-selected 33 loci in two T-ALL lines to Sanger sequence that were near genes recognized to play roles in T-ALL biology or that were relatively long. We agree with the reviewer that these candidates may not represent the overall success rates of our pipeline. To address this, we have randomly selected a total of 103 insertions at two steps of pipeline filtration: 1) 82 at initial prediction of insertions in enhancers and 2) 23 insertions for which ChIP-seq reads were biased towards sequences that contained the insertion. We believe that the read-biasing insertions are more likely to influence enhancer activity, because increased protein binding results in increased read production, and the published *TAL1*-proximal insertion displays these characteristics [1]. In the submitted manuscript, 66% of hand-selected insertions in enhancers were confirmed to be present in the genomes of the cells. With our new data, we can now show that 71% of 68 randomly selected insertions in enhancers were confirmed to be present in tested genomes, and 96% of 23 randomly selected read-biasing insertions are present in the tested genomes.

Furthermore, we had analyzed ChIP-Seq from the GM12878 B lymphoblastoid cell line in the initial submission, for which Illumina “Platinum” whole-genome sequence and variants are also available. We confirmed that 74% of our predicted insertions are also identified by Illumina, and 53% of our insertions predicted to alter enhancers are also identified by Illumina. We have added these analyses to the text on page 4 and page 5:

Four lines of evidence confirmed that the method described here captured bona fide insertions present in tumour genomes. First, we searched for previously validated TAL1-proximal enhancer-associated insertions in MOLT4 and Jurkat T-ALL cells⁸; these were rediscovered in our catalogue (Figure 2A). Second, we subjected a random subset of 68 enhancer-associated insertion candidates in MOLT4 T-ALL cells to high-throughput sequencing, which confirmed that 48 (71%) of the predicted insertions were indeed present in these tumour genomes (Figure 2B, Supplemental Table 2). Third, we carried out targeted Sanger sequencing of 37 candidate loci with insertions in MOLT4, Jurkat, Kelly, SH-SY5Y and LS174T cells, which confirmed the majority of these loci (29 of 37; 78%) contained insertions (Figure 2C, Supplemental Table 3, Supplementary Data 1). Finally, 74% (1,112 of 1,492) predicted enhancer-associated insertions in the GM12878 cell line were supported by an Illumina Platinum Genome sequence (Figure 2D, Supplemental Table 3)³⁸. Confirmed insertions span the size range of the predictions (1-22 bp) and include examples of homozygous and heterozygous insertions. Thirty-six of 48 high-throughput-confirmed insertions and 17 of 29 Sanger-confirmed insertions were found to be heterozygous, which is a feature we noted previously for the TAL1-proximal enhancer-associated insertions, although both homozygous and heterozygous insertions may alter gene expression⁸. These lines of evidence suggest that many of the predicted

enhancer-associated insertions in the catalogue reflect bona fide insertions in tumour cell genomes.

The insertions that bias ChIP-Seq coverage by greater than 2-fold are more likely to be present in the genomes of tumour cells; 96% of 23 randomly selected coverage-biasing insertions were confirmed to be present as either heterozygotes or homozygotes by targeted sequencing in MOLT4 cells (Figure 3C, 3D, 3E).

- Even among the real insertions detected there is still an important issue to solve, if they are somatic or germline variation. A number of detected insertions (33,703) were discarded as they appeared in more than one tumor sample, however an unknown proportion of the ones that appear in only one sample may also be germline variation. Could authors comment on this?

We have directly tested whether the predicted insertions reflect somatic or germline variation using polymorphism data from dbSNP, which curates the 1000 Genomes Project mutations among others. The absence of matched germline samples precludes direct assessment of the origin of these variants. Indeed, our previously discovered, *TAL1*-proximal insertions were somatically acquired, but it is important to note that germline variation can have similar consequences on gene expression. Nevertheless, we have prioritized non-germline variants as most driver variants in the coding genome are somatically acquired.

We calculated that ~30% of all called insertions are present in dbSNP and are likely to thus reflect germline variation. Of those insertions identified in multiple samples, 66% are present in dbSNP. In contrast, insertions specific to one or two samples show only 25% are present in dbSNP. This is consistent with highly recurrent insertions reflecting germline variation, and has also aided in refining our pipeline to increase the likelihood that detected insertions are somatic. We have added a discussion of this to the text on page 4:

Although germline variation may contribute to oncogene misregulation²¹, most cancer-driving variants are somatic in origin, so insertions judged likely to be background, germline variation were eliminated from further study. Of 168,149 candidate enhancer-associated insertions with unique positions and/or sequences, 50,836 were deprioritized because they likely reflect germline variation based on two considerations (Figure 1C, Supplementary Data 1). Of the 57,013 putative germline insertions, 49,992 were present in dbSNP, which curates germline variants of multiple variant types across many databases³⁷. Furthermore, 20,715 variants were recurrent across multiple independent tumour types and samples and may thus reflect germline variation not represented in the reference genome. Indeed, 13,694 of the 20,715 nearly ubiquitous insertions were present in dbSNP³⁷, supporting the view that these insertions are predominantly germline. Thus, 111,136 (168,149 minus 57,013) unique predicted

enhancer-associated insertions appear, by these considerations, not to reflect germline variation and are thus likely somatically acquired non-coding variants (Extended Data Figure 1E).

Furthermore, we have acquired matched patient data for the three of the four patients whose tumors had *LMO2*-proximal insertions. All three of these insertions were somatically acquired. An additional cohort of T-ALL samples contained one sample with an *LMO2*-proximal insertion that, by analysis of diagnosis, remission, and relapse samples, was demonstrated to be somatically acquired. We have added a discussion of this to the text on page 6:

In 3 of 3 patients where non-tumor cells were available, we did not observe the insertion in the non-tumor cells by Sanger sequencing, suggesting that the insertion was somatically acquired in the tumor samples. In an additional cohort of T-ALL samples with whole-genome sequences, we found a 45 bp insertion present at this locus at diagnosis and relapse but absent in remission DNA, suggesting this insertion was somatically acquired (Extended Data Figure 2B).

*- Also, among the 33,703 discarded insertions there may be recurrent oncogenic somatic insertions, although most I agree are probable germline variation. Based on the results from a previous article by the authors (Mansour et al., Science 2014), a recurrent insertion in the enhancer of *TAL1* (exactly the same insertion was found in 3 patients) was shown to be oncogenic. Authors should also discuss this possibility.*

We agree with the reviewer that recurrent and/or germline variation may contribute to oncogene misregulation. In addition to the *TAL1*-proximal case discussed in Mansour et al., we have previously shown that a recurrent inherited variant contributes to neuroblastoma through a similar mechanism of oncogene misregulation [1, 5]. However, given our number of samples per tumor type, and that the functional *TAL1*-proximal was only observed in two of our cell lines, we chose to consider insertions present in greater than two samples as recurrent with a higher probability of being germline in origin. This was supported by our analysis of known germline variants, which showed that a majority of the recurrent insertions present in more than two samples could be germline. In the absence of matched normal samples, this filter is useful in helping discern germline variation. We have clarified this possibility in the text on page 4:

Although germline variation may contribute to oncogene misregulation²¹, most cancer-driving variants are somatic in origin, so insertions judged likely to be background, germline variation were eliminated from further study.

- An unknown number of the detected insertions are real and somatic. Among those probably only very few are oncogenic, as most of those are likely to be passenger insertions. To find the probable oncogenic insertions in enhancers the authors focus on those that occurred within the same insulated neighborhoods as known oncogenes. The rest of the manuscript focus in the detailed analysis of an insertion in an enhancer of LMO2 gene in T-ALL cells.

We agree with the reviewer that it was unclear how many of the insertions are likely to be truly oncogenic. Our pipeline is designed to capture those insertions likely to be functional at the level of gene expression alterations, and we can only suggest that those insertions that affect expression of oncogenes have some increased likelihood of being truly oncogenic. We have added text clarifying this issue on page 5:

To find the subset of insertions in our catalogue that are likely to increase enhancer activity, we filtered for insertions whose presence correlates with increased ChIP-Seq signal for nucleosomes with histone H3K27ac, which occupy active enhancers¹⁴. The TAL1-proximal insertions that cause elevated enhancer activity show such a bias⁸, so we identified insertions that have increased insertion-containing reads relative to reference sequence reads.

Reviewer #2 (Remarks to the Author):

Brian Abraham et al report a strategy designed to identify insertions in non-coding regions of the genome that may contribute to aberrant expression of oncogenes. The manuscript contains what appears to be an intuitive computational pipeline for identifying such insertions, and then also some validation of a regulatory element in the LMO2 gene locus. While the data contained in this manuscript seem to be robust, there are several instances where in my view the study is rather preliminary. Consequently there is little certainty for any potential for wider future impact with the paper as it stands.

Our view is that this approach will prove to be valuable for identifying an important class of somatic variants that contribute to oncogene dysregulation, and thus should have considerable impact. However, we appreciate the reviewer's concerns, and we have addressed these by integrating additional data and analysis, including primary patient data, into the manuscript, as described in more detail below.

Major Points:

1) The actual nature of these enhancer insertions remains unclear. It was impressive but possibly somewhat underhand that the authors managed to write this paper in a way that suggests the mutations are somatically acquired without

ever stating it. And the reason they are not stating it is because they have no evidence for it. But that leaves the question wide open as to what proportion of the so-called insertions are actually polymorphisms, and which ones are acquired mutations that cause ectopic expression of oncogenes. Without any data on this critical point, the paper is inconclusive and very preliminary.

Variation that alters gene expression in tumors need not be somatic, which we have shown in our previous work on inherited variation altering gene expression in neuroblastomas [5]. Nevertheless, we apologize for the lack of clarity in the text when describing the source of the insertions. Our samples lack corresponding normal samples with which we can analyze the origin of variations so we now use other sources of information to identify likely germline variation. We further addressed this concern experimentally. We have acquired substantial evidence that the majority of insertions in our catalogue are not germline variation. We have also acquired matched patient samples and have shown that, similar to *TAL1*-proximal case, the *LMO2*-proximal insertion was acquired somatically in patients. These results are described in more detail below.

Absolutely critical experiments are:

i) Analysis of constitutional/germline DNA from matched tumor/normal samples. I realize this may be difficult for the cell lines, but then they should look at patient samples where normal control DNA would often be available (indeed later in the paper there is talk of using some patient samples)

We have acquired matched patient data for the three of the four patients whose tumors had *LMO2*-proximal insertions. All three of these insertions were somatically acquired. An additional cohort of T-ALL samples contained one sample with an *LMO2*-proximal insertion that, by analysis of diagnosis, remission, and relapse samples, was demonstrated to be somatically acquired. We have added a discussion of this to the text on page 6:

In 3 of 3 patients where non-tumor cells were available, we did not observe the insertion in the non-tumor cells by Sanger sequencing, suggesting that the insertion was somatically acquired in the tumor samples. In an additional cohort of T-ALL samples with whole-genome sequences, we found a 45 bp insertion present at this locus at diagnosis and relapse but absent in remission DNA, suggesting this insertion was somatically acquired (Extended Data Figure 2B).

Re-analysis of whole-genome sequencing data from a T-ALL patient cohort was able to confirm about half of our T-ALL cell line predictions. 15,512 of 38,233 predicted insertions in cell line genomes had a comparable insertion in a T-ALL tumor genome. We have added a discussion of this to the text on page X:

Reanalysis of whole-genome sequences from T-ALL patients showed that half of predictions in T-ALL cell lines exist in patient genomes, demonstrating that whole-genome sequencing is capable of interrogating these variants, but current analysis pipelines commonly discard them.

ii) A thorough investigation of how many of the so-called insertions are polymorphisms. Detailed analysis of for example the 10,000 genome project data needs to be undertaken, as well as other publicly available full genome sequence data collections. It will be possible to screen bioinformatically through the raw data of these collections specifically with the "insertion region sequences" and then see whether or not there are any matches in normal individuals.

We have directly tested whether the predicted insertions reflect somatic or germline variation using polymorphism data from dbSNP, which curates the 1000 Genomes Project mutations among others. The recently published 10,000 genomes from the Venter group were not easily available to us at the time of review. The absence of matched germline samples precludes direct assessment of the origin of these variants. Indeed, our previously discovered, *TAL1*-proximal insertions were somatically acquired, but it is important to note that germline variation can have similar consequences on gene expression. Nevertheless, we have prioritized non-germline variants as most driver variants in the coding genome are somatically acquired.

We calculated that ~30% of all called insertions are present in dbSNP and are likely to thus reflect germline variation. Of those insertions identified in multiple samples, 64% are present in dbSNP. In contrast, insertions specific to one or two samples show only 25% are present in dbSNP. This is consistent with highly recurrent insertions reflecting germline variation, and has also aided in refining our pipeline to increase the likelihood that detected insertions are somatic. We have added a discussion of this to the text on page 4:

Although germline variation may contribute to oncogene misregulation²¹, most cancer-driving variants are somatic in origin, so insertions judged likely to be background, germline variation were eliminated from further study. Of 168,149 candidate enhancer-associated insertions with unique positions and/or sequences, 50,836 were deprioritized because they likely reflect germline variation based on two considerations (Figure 1C, Supplementary Data 1). Of the 57,013 putative germline insertions, 49,992 were present in dbSNP, which curates germline variants of multiple variant types across many databases³⁷. Furthermore, 20,715 variants were recurrent across multiple independent tumour types and samples and may thus reflect germline variation not represented in the reference genome. Indeed, 13,694 of the 20,715 nearly ubiquitous insertions were present in dbSNP³⁷, supporting the view that these insertions are predominantly germline. Thus, 111,136 (168,149 minus 57,013) unique predicted

enhancer-associated insertions appear, by these considerations, not to reflect germline variation and are thus likely somatically acquired non-coding variants (Extended Data Figure 1E).

2) It was unclear to me why some of the so-called insertions were homozygous. For a "gain of function" event such as the one they imply for LMO2, alteration of a single allele should be enough. Therefore, what do the authors think is going on in the homozygous samples? Is it loss of function leading to loss of expression of a tumour suppressor? But then why would this be picked up with acetylated histones? Or is it acquired homozygosity through mitotic recombination? Or is it indeed a polymorphism that happens to be homozygous in this particular sample (see point above in relation to what proportion of the events are actually polymorphisms).

The reviewer suggests several plausible models for homozygous insertions being detected by our pipeline. In all models, however, the reference genome does not represent the sequence in these cells. Homozygous insertions can be acquired through uniparental disomy, wherein cells require two hits for sufficient levels of a gene product [6], loss of a wild-type allele through aneuploidy, or inherited homozygosity from both parental alleles. Indeed, 8 of 31 confirmed homozygotes are contained in dbSNP and thus likely represent germline variants. Despite the varied sources, some of these homozygous insertions may still affect enhancer activity and gene expression, albeit similarly on both alleles. To address this range of models and consequences, we have added the following text to the manuscript on page 5:

Thirty-six of 48 high-throughput-confirmed insertions and 17 of 29 Sanger-confirmed insertions were found to be heterozygous, which is a feature we noted previously for the TAL1-proximal enhancer-associated insertions, although both homozygous and heterozygous insertions may alter gene expression⁸.

3) What is the background mutation rate in these cultured cancer cell lines? Cancer cells commonly have defects in DNA repair, and cancer cell lines often have very messed-up genomes. How can the authors exclude that such in vitro processes are not major contributors to what they describe?

The mutation rates in cultured cancer cell lines varies, but estimates place the rates between 2×10^{-7} mutations/cell/generation to 30×10^{-7} mutations/cell/division, so, while we appreciate the reviewer's concern about cell culture as a convoluting factor in interpreting our insertions, we believe this to be a negligible contributor [7, 8]. Despite this relatively low rate, insertions occurring in cultured tumor cells can still contribute to oncogene misregulation, so we can compare cell line samples with the patient samples in our initial submission. We identified comparable numbers of insertions in primary patient ChIP-Seq samples as cell

lines in our catalogue. The mean numbers of enhancer-associated insertions per sample were 2283 and 3313 for primary samples and cell lines, respectively, and the Student's T test p-value comparing the distributions of these counts was 0.08, indicating no significant difference in number. Furthermore, analysis of non-cancer cultured cell lines identifies substantially fewer non-germline insertions in GM12878 (233) and IMR90 (225) per sample than the cancer cell lines, consistent with culturing itself being a smaller contributor than the tumor state itself.

4) It was difficult to assess which Chip-seq experiments were newly done for this paper, and which were old (bottom of page 3). Why not just be upfront and give the actual numbers?

We have modified Supplemental Table 1 to clarify which datasets are new in this publication. The GEO identifiers have been included. We also now mention the number of newly created samples in the text on page 3:

The pipeline was used to analyse newly generated and previously published ChIP-Seq datasets for H3K27ac-enriched DNA from 78 tumour cell lines and 24 primary tumour samples, 8 of which are new here

5) A quick scan of the literature suggests that a fair amount is known about the regulation of LMO2. Does the region picked up in this paper correspond to any of the previously mapped LMO2 regulatory regions? Is it therefore perhaps a known element with added ectopic activity, or are the authors suggesting generation of a completely new element de novo?

To our knowledge, this is the first time an active enhancer was noted at this genomic site upstream of *LMO2*. The region does not appear to be used in thymocytes or other T-ALLs. The *LMO2*-proximal insertion occurs in a region that has nearby DNA sequences predicted to bind other T-ALL regulators such as T-BET and NFKB1. This is consistent with our previous study on a *TAL1*-proximal insertion, wherein a MYB-binding sequence was created adjacent *TAL1* and *GATA3* sequence motifs. Furthermore, the *LMO2*-proximal enhancer seen in the MOLT-4 T-ALL cell line is near regulatory elements that are active in other blood cell types. CD34 hematopoietic stem and progenitor cells have an active enhancer in the region nearby the MOLT-4 enhancer. Together, this suggests that there was some potential for transcription factors to bind the DNA where the *LMO2*-proximal insertion occurred, but it was not used in thymocytes or other T-ALLs. We have added a figure panel showing the *LMO2*-proximal insertion locus in other, non-T-ALL cell types to Extended Data Figure 2A and a discussion of this potentiation to the text on page 6:

Among the T-ALL cells studied here, both the insertion and an active enhancer at this region were unique to MOLT4 cells (Figure 4C). The location of this novel enhancer is not consistent with simple reactivation of

a developmental enhancer (Extended Data Figure 2A).

6) Table 1: Analysis needs to be performed to determine whether the number of oncogenes observed is more than one would expect by chance. If not, the table may be meaningless.

There are significantly more observed insertions in the neighborhoods of oncogenes than predicted by chance ($p < 0.0001$). We performed an analysis of whether randomly generated mutations are in the insulated neighborhoods containing oncogenes. We have added this to the text on page 6:

Many notable oncogenes occur in the same insulated neighborhoods as enhancer-associated insertions that affect enhancer signal (Table 1). Indeed there was a significant enrichment of enhancer-associated insertions in insulated neighborhoods that contain oncogenes ($p < 0.0001$).

Furthermore, the assignment of insertions to their potential target genes is of general interest to labs wishing to study the regulation and misregulation of oncogenes.

1. Mansour MR, Abraham BJ, Anders L, Berezovskaya A, Gutierrez A, Durbin AD, Etchin J, Lawton L, Sallan SE, Silverman LB, Loh ML, Hunger SP, Sanda T, Young RA, Look AT: **An oncogenic super-enhancer formed through somatic mutation of a noncoding intergenic element.** *Science* 2014, **346**:1373–7.
2. Levy S, Sutton G, Ng PC, Feuk L, Halpern AL, Walenz BP, Axelrod N, Huang J, Kirkness EF, Denisov G, Lin Y, MacDonald JR, Pang AWC, Shago M, Stockwell TB, Tsiamouri A, Bafna V, Bansal V, Kravitz S a., Busam D a., Beeson KY, McIntosh TC, Remington K a., Abril JF, Gill J, Borman J, Rogers YH, Frazier ME, Scherer SW, Strausberg RL, et al.: **The diploid genome sequence of an individual human.** *PLoS Biol* 2007, **5**:2113–2144.
3. Eberle MA, Fritzilas E, Krusche P, Kallberg M, Moore BL, Bekritsky MA, Iqbal Z, Chuang H-Y, Humphray SJ, Halpern AL, Kruglyak S, Margulies EH, McVean G, Bentley DR: *A Reference Dataset of 5.4 Million Human Variants Validated by Genetic Inheritance from Sequencing a Three-Generation 17-Member Pedigree.* Cold Spring Harbor Labs Journals; 2016.
4. Sherry ST, Ward MH, Kholodov M, Baker J, Phan L, Smigielski EM, Sirotkin K: **dbSNP: the NCBI database of genetic variation.** *Nucleic Acids Res* 2001, **29**:308–11.
5. Oldridge DA, Wood AC, Weichert-Leahey N, Crimmins I, Sussman R, Winter C, McDaniel LD, Diamond M, Hart LS, Zhu S, Durbin AD, Abraham BJ, Anders L, Tian L, Zhang S, Wei JS, Khan J, Bramlett K, Rahman N, Capasso M, Iolascon A, Gerhard DS, Guidry Auvil JM, Young RA, Hakonarson H, Diskin SJ, Look AT,

- Maris JM: **Genetic predisposition to neuroblastoma mediated by a LMO1 super-enhancer polymorphism.** *Nature* 2015, **528**:418–21.
6. Raghavan M, Smith L-L, Lillington DM, Chaplin T, Kakkas I, Molloy G, Chelala C, Cazier J-B, Cavenagh JD, Fitzgibbon J, Lister TA, Young BD: **Segmental uniparental disomy is a commonly acquired genetic event in relapsed acute myeloid leukemia.** *Blood* 2008, **112**:814–821.
7. BOESEN J, NIERICKER M, DIETEREN N, SIMONS J: **How variable is an spontaneous mutation rate in cultured mammalian cells?** *Mutat Res Mol Mech Mutagen* 1994, **307**:121–129.
8. Araten DJ, Golde DW, Zhang RH, Thaler HT, Gargiulo L, Notaro R, Luzzatto L: **A quantitative measurement of the human somatic mutation rate.** *Cancer Res* 2005, **65**:8111–7.

Reviewers' comments:

Reviewer #2 (Remarks to the Author):

First of all, it is really annoying not to have the revisions marked in red font (or underlined) in the revised text. It is therefore hard for me to judge what exact bits of text have been changed. Saying that, the specific responses in the rebuttal letter to my comments seem OK, at first hand at least. The one issue though on which I got stuck was the response to my comment 5. I felt the response that there was no need on my part to worry about what may be known about this region near LMO2 was a bit glib, and I therefore had a look at the region (finding it via the sequence given in extended data figure 2B). I have e-mailed a screenshot of the relevant UCSC browser to the editor so that they can make up their own mind. The two issues I noted were:

1) The region that is reported to have all these insertions is in a repeat. I could not see any discussion of this in the text, although it may have implications for frequency of polymorphisms, mutations as well as issues to do with the ChIP-Seq (mapping accuracy for example).

2) Much more concerning, the very region is annotated by the ENCODE consortium as a region of a DNaseI Hypersensitive Site Peak Cluster, and indeed they report this to be open chromatin in multiple cell lines as well as primary cultured human dermal endothelial cells. This to me seems at complete odds with the model put forward here that oncogenic insertions, somatically acquired, generate an enhancer where ordinarily there is nothing, and this causes ectopic activation of LMO2.

I can see that the overall concept of the paper is attractive and newsworthy, but am not convinced that the proposed mechanistic aspects around the LMO2 expression are nearly as straightforward as they are purported to be.

Reviewer #3 (Remarks to the Author):

Abraham et al perform an analysis of enhancer-associated small insertions in 102 cancer genomes. To this end the authors employ a ChIP-Seq enrichment strategy targeting H3K27ac to enrich for active enhancer elements. The authors employ a bioinformatics filtering technique to remove known germline variation from their data. After extensive filtering the authors report 111,136 enhancer-associated variants. In the second part of the manuscript the authors study enhancer-associated insertions proximal to the LMO2 gene in detail and demonstrate that the variant allele carrying an 8bp insertion has higher levels of H3K27ac and gene expression.

The study is interesting and timely. The manuscript is well written and Figures are clear and informative.

As a major criticism I agree with reviewer 1 that it remains unclear to which extent the described indels are somatic. Although the authors have employed substantial filtering using public data bases and recurrence across their unrelated cancer samples this remains an unanswered question in the absence of normal controls. Hence no strong claims about the somatic origin should be made.

The authors describe a total of 20,715 novel germline insertions, observed in more than 1 tumour genome, but absent in data bases of germline variation. Given the small sample size (102) it seems very likely that this figure would increase substantially if more genomes would be sequenced. Due to the technical challenges of indel calling it is likely that databases of genome variation are depleted for this type of polymorphisms.

Another warning sign is the observation that only 17 of 29 Sanger-confirmed variations were heterozygous. Hence at least 12 are very likely to be germline variants as somatic mutations occur on only one allele and only become homozygous if there is additional loss of heterozygosity.

Please see below a point-by-point response to the reviewers' concerns.

Reviewer #2

First of all, it is really annoying not to have the revisions marked in red font (or underlined) in the revised text. It is therefore hard for me to judge what exact bits of text have been changed. Saying that, the specific responses in the rebuttal letter to my comments seem OK, at first hand at least. The one issue though on which I got stuck was the response to my comment 5. I felt the response that there was no need on my part to worry about what may be known about this region near LMO2 was a bit glib, and I therefore had a look at the region (finding it via the sequence given in extended data figure 2B). I have e-mailed a screenshot of the relevant UCSC browser to the editor so that they can make up their own mind.

The analysis we carried out led us to conclude that the *LMO2*-proximal insertion creates an enhancer that is not used in other healthy blood cells. This is consistent with the model that an insertion within a region with unused binding sites for additional transcription factors can sometimes nucleate an enhancer. Below we provide a more thorough explanation for how the data provided in the manuscript and response letter support our model for the *LMO2*-proximal insertion and describe modifications to the text to clarify the issue.

The two issues I noted were:

1) The region that is reported to have all these insertions is in a repeat. I could not see any discussion of this in the text, although it may have implications for frequency of polymorphisms, mutations as well as issues to do with the ChIP-Seq (mapping accuracy for example).

The *LMO2*-proximal insertion occurs within a low-fidelity SINE repeat element. This particular region is so divergent from the canonical SINE element that our computational and experimental assays can confidently assign the insertion-containing sequence to this single site.

Two lines of evidence argue that the genomic position of this insertion was correctly identified. First, the computational analysis identifies this position as the only likely source of this sequence. The 53-nucleotide contig within which the *LMO2*-proximal insertion was computationally identified was built with 5x coverage of each base, which rules out the possibility that the presence of this sequence is a technical artifact. We used multiple alignment tools (BLAT, Bowtie2) that require unique mapping positions, and both of these tools identified only one unique genomic position matching the whole contig. Second, the PCR primers we used to amplify and Sanger-sequence this insertion in genomic DNA uniquely recognize a single genomic locus. One of the primers anneals outside the predicted repeat element, and 50% of the sequence of the second primer binds outside the repeat element. The primer pair produced a single PCR amplicon with flanking sequence outside the insertion sequence, indicating that we interrogated a unique position in the genome. We are thus confident that we are interrogating this exact position in the human genome.

2) Much more concerning, the very region is annotated by the ENCODE consortium as a region of a DNase I Hypersensitive Site Peak Cluster, and indeed they report this to be open chromatin in multiple cell lines as well as primary cultured human dermal endothelial cells. This to me seems at complete odds with the model put forward here that oncogenic insertions, somatically acquired, generate an enhancer where ordinarily there is nothing, and this causes ectopic activation of *LMO2*.

We cannot find compelling evidence for enhancer activity in the region containing the *LMO2*-proximal insertion in normal human cells. The reviewer is correct that there is some DNase I hypersensitivity signal from ENCODE data in some cultured cells, including primary cultured human dermal endothelial cells, but careful examination shows that the signal for DNase I is very weak in all but one sample, Jurkat T-ALL. Furthermore, the presence of DNase I hypersensitivity is not necessarily indicative of an active enhancer, as these sites can also mark silencers and insulators.

I can see that the overall concept of the paper is attractive and newsworthy, but am not convinced that the proposed mechanistic aspects around the *LMO2* expression are nearly as straightforward as they are purported to be.

We respectfully submit that the computational and experimental analysis strongly supports the proposed model. To determine whether the 8 bp heterozygous insertion in the *LMO2* locus confers enhancer activity, the insertion allele and the reference allele were cloned into enhancer reporter vectors, and these were transfected into Jurkat T-ALL cells; the results indicate that the insertion allele has significantly more enhancer activity than the reference allele (Figure 4E). Among the T-ALL cells studied here, both the insertion and an active enhancer at this region were unique to MOLT4 cells (Figure 4C).

The location of this aberrant enhancer is not consistent with simple reactivation of a developmental enhancer (Extended Data Figure 2A). In normal double-positive thymocytes, there is no evidence for an active enhancer at the region containing the *LMO2*-proximal insertion. *LMO2* is expressed in blood progenitors, but expression is mediated by a separate, developmentally regulated enhancer¹. Expression of *LMO2* must be downregulated developmentally or a multi-step cascade leading to T-ALL will be induced.

We now address these points in the text: “The insertion falls in a predicted SINE repeat occurrence that is uniquely mappable, so the insertion could be uniquely localized to this site using ChIP-Seq reads^{2,3}. *LMO2* is not expressed in normal mature thymocytes, and its aberrant expression in these cells is thought to initiate a series of events leading to leukaemia¹, so its misregulation by enhancers is of particular interest. Interestingly, a DNase I-hypersensitive site is present at the locus in a related T-ALL cell line, Jurkat, suggesting the region has a possibility of acting as an enhancer⁴, despite DNase I signal alone not necessarily representing an active enhancer⁵. Together, these data suggest the insertion is near a region with potential regulatory capacity near a key oncogene in leukaemia.”

“The location of this aberrant enhancer is not consistent with simple reactivation of a developmental enhancer (Supplementary Fig. 2A), which is a proposed general phenomenon explaining oncogenic enhancer activation^{6,7}.”

Reviewer #3

Abraham et al perform an analysis of enhancer-associated small insertions in 102 cancer genomes. To this end the authors employ a ChIP-Seq enrichment strategy targeting H3K27ac to enrich for active enhancer elements. The authors employ a bioinformatics filtering technique to remove known germline variation from their data. After extensive filtering the authors report 111,136 enhancer-associated variants. In the second part of the manuscript the authors study enhancer-associated insertions proximal to the *LMO2* gene in detail and demonstrate that the variant allele carrying an 8bp insertion has higher levels of H3K27ac and gene expression.

The study is interesting and timely. The manuscript is well written and Figures are clear and informative.

As a major criticism I agree with reviewer 1 that it remains unclear to which extent the described indels are somatic. Although the authors have employed substantial filtering using public data bases and recurrence across their unrelated cancer samples this remains an unanswered question in the absence of normal controls. Hence no strong claims about the somatic origin should be made.

We agree with the reviewer that, because our study was done in human tumor cell lines, we cannot definitely state whether any of the insertions detected in these lines occurred somatically in the development of the cancer. This is because, in the era in which these cell lines were established, corresponding germline samples were not collected. We note in the paper that we used filters to exclude known polymorphisms to enrich as much as possible for abnormalities that represent somatic events and were selected to promote the malignant phenotype. Nonetheless, we have modified the language in the manuscript to make no strong claims about somatic origin unless this is demonstrated. For those insertions where we explicitly note somatic origin, we do have access to comparable normal/tumor data and have shown that these are acquired insertions. In the revised manuscript, we have explicitly noted somatic origin for samples when we have access to matched normal DNA from the same patient and have shown that the insertion is acquired.

We note that it not necessary for insertions functional at the level of gene misregulation to be somatically acquired, so it is of interest to have a catalogue that include both germline and somatic insertions. Indeed, a known germline variant contributes to neuroblastoma via a similar mechanism of gene misregulation⁸.

The authors describe a total of 20,715 novel germline insertions, observed in more than 1 tumour genome, but absent in data bases of germline variation. Given the small sample size (102) it seem very likely that this figure would increase substantially if more genomes would be sequenced. Due to the technical challenges of indel calling it is likely that databases of genome variation are depleted for this type of polymorphisms.

We agree that small insertions outside the protein-coding genome are underrepresented in variant databases. This was one motivation for using an additional filter of cross-sample presence to identify variants likely to be germline. As a small point of clarification, we required the insertion to be present in more than two samples to be considered germline.

Another warning sign is the observation that only 17 of 29 Sanger-confirmed variations were heterozygous. Hence at least 12 are very likely to be germline variants as somatic mutations occur on only one allele and only become homozygous if there is additional loss of heterozygosity.

We agree that it is possible our confirmed homozygous mutations are germline in nature. Pathogenic biallelic insertions can be selected for by uniparental isodisomy or loss of the normal allele, in either case leading to loss of heterozygosity. It is possible that homozygous mutations have a similar effect on gene expression, either from the retained mutant allele or from both alleles in the case of uniparental isodisomy^{9,10}.

We have added further discussion of this to the text: “The catalogue reported here includes enhancer-associated insertions in T-ALL, breast, neuroblastoma (NB), lung, colorectal, melanoma, glioblastoma multiforme (GBM), B cell lymphoma (BCL), pancreatic, and other tumour cell types (Supplementary Data 1). This catalogue likely contains both germline and somatically acquired variants. Although most cancer-driving variants described thus far are somatic in origin, and somatic variants are considered more likely to be functionally important than germline variants, the somatic or germline origin of most variants described here could not be determined for the tumor cells in this study. Nonetheless, germline variants in noncoding DNA may contribute to transcriptional dysregulation of tumor oncogenes⁸, so it is useful to have the a catalogue of both types of enhancer-associated variants.”

We have also modified the text: “Thus, 111,136 (168,149 minus 57,013) unique predicted enhancer-associated insertions appear, by these considerations, not to reflect germline variation and thus may be somatically acquired non-coding variants (Supplementary Fig. 1E).”

1. Matthews, J. M., Lester, K., Joseph, S. & Curtis, D. J. LIM-domain-only proteins in cancer. *Nat. Rev. Cancer* **13**, 111–122 (2013).
2. Smit, A., Hubley, R. & Green, P. RepeatMasker Open-3.0. at <<http://www.repeatmasker.org>>

3. Derrien, T. *et al.* Fast Computation and Applications of Genome Mappability. *PLoS One* **7**, e30377 (2012).
4. ENCODE Project Consortium. An integrated encyclopedia of DNA elements in the human genome. *Nature* **489**, 57–74 (2012).
5. Gross, D. S. & Garrard, W. T. Nuclease Hypersensitive Sites in Chromatin. *Annu. Rev. Biochem.* **57**, 159–197 (1988).
6. Herranz, D. *et al.* A NOTCH1-driven MYC enhancer promotes T cell development, transformation and acute lymphoblastic leukemia. *Nat. Med.* **20**, (2014).
7. Sur, I. & Taipale, J. The role of enhancers in cancer. *Nat. Rev. Cancer* (2016). doi:10.1038/nrc.2016.62
8. Oldridge, D. A. *et al.* Genetic predisposition to neuroblastoma mediated by a LMO1 super-enhancer polymorphism. *Nature* **528**, 418–21 (2015).
9. Ferrando, A. A. *et al.* Biallelic transcriptional activation of oncogenic transcription factors in T-cell acute lymphoblastic leukemia. *Blood* **103**, 1909–11 (2004).
10. Raghavan, M., Gupta, M., Molloy, G., Chaplin, T. & Young, B. D. Mitotic recombination in haematological malignancy. *Adv. Enzyme Regul.* **50**, 96–103 (2010).

REVIEWERS' COMMENTS:

Reviewer #3 (Remarks to the Author):

The authors have addressed my previous concerns about the somatic nature of indels and are now using a more cautious wording.

I have no other concerns.

Moritz Gerstung

European Bioinformatics Institute EMBL-EBI